# Contribution of the medial eye field network to the voluntary deployment of visuospatial attention

Guillaume Herbet [1,2 ✉] & Hugues Duffau [1,2]

Historically, the study of patients with spatial neglect has provided fundamental insights into the neural basis of spatial attention. However, lesion mapping studies have been unsuccessful in establishing the potential role of associative networks spreading on the dorsal-medial axis, mainly because they are uncommonly targeted by vascular injuries. Here we combine machine learning-based lesion-symptom mapping, disconnection analyses and the longitudinal behavioral data of 128 patients with well-delineated surgical resections. The analyses show that surgical resections in a location compatible with both the supplementary and the cingulate eye fields, and disrupting the dorsal-medial fiber network, are specifically associated with severely diminished performance on a visual search task (i.e., visuo-motor exploratory neglect) with intact performance on a task probing the perceptual component of neglect. This general finding provides causal evidence for a role of the frontal-medial network in the voluntary deployment of visuo-spatial attention.

[1] Institute of Functional Genomics, University of Montpellier, INSERM U1191, CNRS UMR 5203, 141, rue de la Cardonille, 34094 Montpellier, France. [2] Department of Neurosurgery, Montpellier University Medical Center, Gui de Chauliac Hospital, 80, Boulevard Augustin Fliche, 34095 Montpellier, France. ✉email: guillaume.herbet@gmail.com

Orienting visual attention towards salient or relevant elements of the surrounding and fast-changing environment is a vital process for adaptive behaviors and survival, especially in species where most behaviors require visual guidance. In humans, accumulating evidence from neuroimaging, behavioral and lesion studies have led to the undisputed view that visuo-spatial attention is maintained through the coordinated activity of networks spreading on the dorsal face of the brain[1], with a central engagement of the fronto-parietal connectivity. Unified models of visuo-spatial attention assume that two anatomically segregated networks are specialized in distinct attentional subprocesses[2–4]. In particular, the dorsal attention network (DAN), mainly composed of the frontal eye field (FEF) and the intraparietal sulcus, may subserve the ability to purposely allocate attention to meaningful elements of the visual scene (top-down, goal-directed orientation), whereas the ventral attention network (VAN), composed of the ventro-lateral prefrontal cortex and the temporo-parietal junction, may be engaged when an unexpected but behaviorally relevant event occurs and attention must be reoriented towards the new visual target (bottom-up, stimuli-driven orientation)—though this dual-pathway anatomo-functional organization would be less marked than previously thought[5,6]. Context-sensitive integration across these two attention networks is hypothesized to allow the dynamic and flexible control of visuo-spatial attention[4]. Beyond this holistic view of attention processing, however, a great deal of uncertainty remains about the exact role of "satellite" but potentially relevant areas in the voluntary deployment of visuo-spatial attention, in particular those lodged in the medial sector of the brain. This is typically the case of the supplementary eye field[7] (SEF) and the cingulate eye field[8] (CEF), two cortical nodes being integrated components of the complex neural circuitry involved in the initiation and the regulation of visually-guided behaviors[9,10], but to date poorly characterized in humans compared to their sister area, i.e., the FEF.

In view of its specific cytoarchitectonic, the FEF has long been considered as strictly involved in the visuo-motor aspect of oculomotor control, including preparation, initiation and execution of saccades[11,12]. This view was initially supported by repeated evidence that low-intensity electrostimulation of the FEF is capable of producing contralateral eye movements in humans[13–16] and primates[17–19], and further bolstered by convergent fMRI findings showing the recurrent activity of this area in various forms of oculomotor activities, including saccadic eye movements[10,12]. However, it has become increasingly accepted that the FEF serves as an interfacing area between the visuo-motor and visuo-spatial attention systems, as it is activated (together with the SEF and the CEF) by saccadic eye movements and by visually triggered overt and covert shifts of attention as well[20]. In addition, the FEF has dense and reciprocal anatomical connections with both functional systems, a connective pattern that is largely shared with the SEF and CEF[21,22]. It is also established that these three anterior eye-related areas are closely interconnected with each other with bidirectional fibers[21,22]. Overall, the FEF is assumed to play a central role in the voluntary control of eye movement and attention towards the contralateral side. The respective role of the SEF and the CEF, however, clearly remains an open question. Current hypotheses suggest the SEF as possibly involved in the higher cognitive aspects of oculomotor behaviors, whereas the CEF would rather be implicated in the motivational aspects of visually-guided actions[9,10]. However, little is known about the contribution of these medial areas to the voluntary control of visual attention, especially in humans. Accordingly, the main goal of the current study was to examine the extent to which damage to the SEF and the CEF as well as

their connective inputs could affect the process of deploying attention towards the contralesional space.

The exploration of patients with spatial neglect (i.e., the debilitating condition commonly observed following a right-sided brain injury and whereby patients show difficulties in orienting attention towards the contralesional space) has been a central catalyst for research on the neural basis of visuo-spatial attention. From available qualitative and quantitative meta-analyses, it clearly emerges that spatial neglect can result from damage to a large set of cortical areas (e.g., temporo-parietal junction, inferior frontal gyrus, and so on) mainly lodged in the dorsal system of the brain or from injuries to subcortical structures such as the thalamus and the caudate nucleus[23]. It is also established that disconnective breakdown of dorsal white-matter tracts, especially the layers II and III of the superior longitudinal fasciculus (SLF) has a detrimental effect on the functional integration within the attention network[24,25] and is associated with the most severe forms of neglect[26–28]. Collectively, these observations have supported the emerging view that spatial neglect may be better conceptualized as stemming from a brain-wide network dysfunction[24,29–34]. One of the accepted shortcomings of lesion mapping studies, however, lies in their inability to fully gauge the entire set of structures that may contribute to a deficit in general, and to spatial neglect in particular, mainly because of the typical non-random distribution of vascular damage[35,36]. As the dominating lesion model has been invariably stroke injury, neuropsychological studies have faced difficulty in delineating the exact role of the dorsomedial structures, which are uncommonly targeted by this particular pathophysiological condition[37].

To assess the potential role of the right medial network in visuo-spatial attention, we relied on the behavioral data of a large cohort of patients presenting with a lower-grade glioma – a rare cerebral tumor mainly characterized by a slow-growth kinetics and preferentially spreading along the white-matter connectivity[38–40]. Over alternate lesion models, the advantage of this particular one is threefold: (i) slow-growing tumors frequently affect the fronto-medial structures[40]; (ii) patients can be evaluated longitudinally before, immediately after and a while after surgery, allowing to gauge both the immediate and longer-term neuropsychological effects of surgical excisions[41]; (iii) the removed structures can be clearly delineated on anatomical MRIs. To assess spatial neglect, two well-tried behavioral paradigms were employed, both of which have been previously associated with task-specific lesion patterns[42–44]. The line bisection task allows assessing how symmetrical is perceived a visual scene, thus probing the "perceptual" aspect of neglect (i.e., representational/perceptive neglect)[34,43]. The related performances are predominantly affected by parietal lesions[44,45]. By contrast, the cancellation task requires patients to voluntarily explore and thus orient attention towards the contralesional space to reach pre-specified items among distractors[42]. Performances on this type of visual search task have been shown to be affected by lesions damaging the DAN, especially at the level of the FEF[42,44,46]. Accordingly, if the medial network is implicated in the voluntary deployment of attention, then surgical excisions of structures shaping it (in particular the SEF and CEF) are expected to be associated with an impaired ability to identify targets in the contralesional space with spared performances on a line bisection task (i.e., visuo-motor exploratory neglect). However, a strict double dissociation is unlikely because target cancellation performances can be affected by perceptive neglect.

To test this straightforward hypothesis, we first used support vector regression lesion-symptom mapping (SVR-LSM)[47,48], a multivariate evolution of the standard mass-univariate lesion mapping approach developed earlier[49] and recently applied to the

study of spatial neglect[50]. SVR-LSM represents an interesting option as the method takes into consideration the interdependent nature of the voxels forming a given lesion map. By this means, we were able to capture a powerful but transient (i.e., circumscribed to the early postoperative period) statistical association between surgical resections targeting both the SEF and the CEF and performances on the cancellation task, not on the line bisection task. The latter one was uniquely affected by parietal resections, as previously shown[43]. We further replicated this pattern of results in an exceptionally rare patient who benefited from a two-step sequential surgery to remove a tumor-infiltrating the right cingulum, involving first the pre-SMA and the caudal part of the anterior cingulate (the typical locations of SEF and CEF, respectively) and 6 months later the right superior parietal lobule and the neighboring precuneus.

In the second line of analyses, we were interested in precisely identifying the role of disconnective disruptions in the emergence of spatial neglect. Current anatomo-functional models of visuospatial attention suggest that layers III and I of the SLF may provide white-matter connectivity for the VAN and the DAN, respectively, whereas layer II may allow both attention systems to communicate[29]. While the association between SLF II and III and spatial neglect has been experimentally evidenced[27,28,51], the exact contribution of SLF_I remains to be clearly elucidated. In view of its known cortical projections in medial frontal areas, we assumed that resective interruption of this tract should impair visual exploration of the contralesional space. To ascertain the extent to which cerebral disconnection predicted behavioral outputs, we used a novel method of estimating the number of fibers interrupted by the operative procedure, based on the population-averaged diffusion data of the Human Connectome Project (HCP). The results confirmed the central role of the SLF I in spatial neglect but also showed similarities and dissimilarities in the disconnectivity patterns associated with impoverished performances in each task.

Overall, the current work shows that surgical resections in a location compatible with both the supplementary and the cingulate eye fields, and damaging the dorsal-medial white-matter network, are specifically associated with severely diminished performance on a visual search task with intact performance on a task probing the perceptual component of neglect. This suggests that the medial eye field network contributes to the voluntary deployment of visuo-spatial attention.

## Results

**Patient sample and background analyses**. Background demographic and clinical variables are fully described in Supplementary Table 1. In brief, the patient sample consisted of 128 patients (mean age: $39.7 \pm 12.3$, 54 females; 121 right-handed) consecutively operated on for a lower-grade glioma (see "Methods" for details about inclusion and exclusion criteria). They were behaviorally assessed at three-time points: the day before surgery (hereafter, A1), 4 days after surgery (hereafter, A2), and 3 months after surgery (hereafter, A3). The average preoperative volume of tumors was $57.5 \text{ cm}^3 \pm 49.0$, whereas the average volume of postoperative resection cavities was $47.2 \text{ cm}^3 \pm 39.7$. Simple correlation analyses indicated that the behavioral measurements of visuo-spatial attention, including line bisection estimates, the total number of omitted bells (Hereafter, total_bell), and the asymmetry score left minus right bells (Hereafter, diff_bell), were poorly associated with the demographic and clinical variables (Supplementary Table 2). This was true for the preoperative level of performance (A1) but also for Δ1 (i.e., the behavioral difference between A1 and A2) and Δ2 (i.e., the behavioral difference between A1 and A3). As a consequence, the variance associated

with these variables was not regressed out from the behavioral measures of interest in the subsequent lesion-symptom analyses.

**Lesions distributions**. Consistent with the typical location of lower-grade glioma[40], the spatial distribution of both pre-operative tumors and resection cavities was rather inhomogeneous across the brain. As expected, the maximum density in the tumor overlap map occurred in the insula extending to the neighboring ventral white-matter connectivity (Fig. 1a). Accordingly, the insula and the adjacent cortical structures, including the temporal pole and the inferior frontal gyrus, were more frequently the target of surgical excisions (Fig. 1b). The dorsal and medial aspects of the frontal cortex, including the pre-supplementary motor area, were also commonly affected by the surgical procedure. Importantly, the FEF (i.e., the intersection between the precentral and superior frontal sulci) was not enough covered by surgical resections to be appropriately taken into consideration in the subsequent lesion-deficit analyses. In keeping with the outcomes of 'awake' surgery with stimulation mapping, structures with low potential for neuroplasticity, including most of the precentral and retrocentral gyri as well as large parts of both the fronto-temporal and the fronto-parietal white-matter connectivity, were only partially removed for functional reasons[40,52].

**Behavioral analyses: patients versus healthy control participants**. To determine whether patients already showed a right bias in visuo-spatial attention before surgeries were performed, pre-operative baseline performances (A1) were statistically compared to those gained from a healthy control group (44 participants) matched in terms of age ($t_{(170)} = -1.13$, $p = 0.26$; two-sided; 95% CI $[-6.19, 1.67]$), educational attainment ($t_{(170)} = 1.06$, $p = 0.29$; two-sided; 95% CI $[-0.44, 1.49]$) and sex ($\chi^2 = 0.14$, $p = 0.71$; two-sided) (see Supplementary Tables 3 and 4 for details about the behavioral data of both the patient group and the control group, respectively). A group effect was observed for line bisection estimates ($t_{(170)} = -3.45$, $p < 0.001$; two-sided; 95% CI $[-2.25, -0.61]$), total_bell ($t_{(170)} = -2.32$, $p = 0.02$; two-sided; CI 95% $[-1.79, -0.15]$) but not for diff_bell ($t_{(170)} = 0.48$, $p = 0.63$; two-sided; 95% CI $[-0.37, 0.61]$). The same analyses were performed considering A2 and A3. In brief, all behavioral measurements were strongly different between groups at A2, and the same pattern than that observed at baseline was identified at A3. All statistical analyses are fully described in Supplementary Table 5 and Supplementary Fig. 1. The frequency of individual deficits estimated from the normative distributions are shown in Supplementary Fig. 2.

**Behavioral analyses: longitudinal performances**. To capture the effect of surgery on behavioral measurements, we statistically assessed how performances evolved over time. To this end, a first repeated measures (RM) ANOVA was performed on the line bisection performance, using assessment time {A1, A2, A3} as the main factor. The results showed that behavior significantly differed across the three measures ($F_{(2, 254)} = 14.7$, $p < 0.001$, $\eta^2_p = 0.10$; two-sided). Pairwise, post-hoc analyses conducted with the Scheffé test revealed that average estimates of line centers significantly shifted toward the right side just after surgery (A1 vs A2; mean $-0.98 \pm 2.25$ vs $0.93 \pm 5.70$; $p < 0.001$; two-sided; 95% CI $[-2.88, -0.96]$), but this effect was only transitory (A2 vs A3; mean $0.93 \pm 5.70$ vs $-0.81 \pm 3.42$, $p < 0.001$; two-sided; 95% CI $[0.78, 2.70]$), so that preoperative and 3-month postoperative estimates did not differ ($p = 0.90$; two-sided; 95% CI $[-1.14, -0.79]$) (Fig. 2a). This was confirmed by a simple, paired t-test

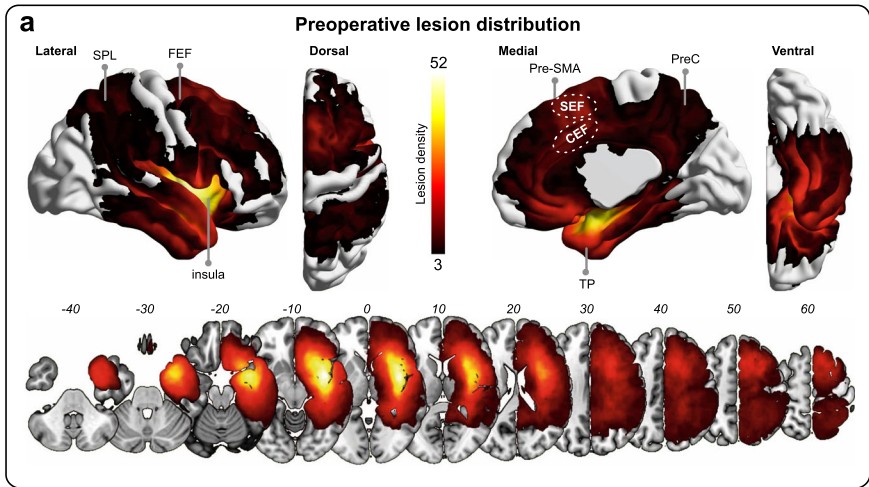

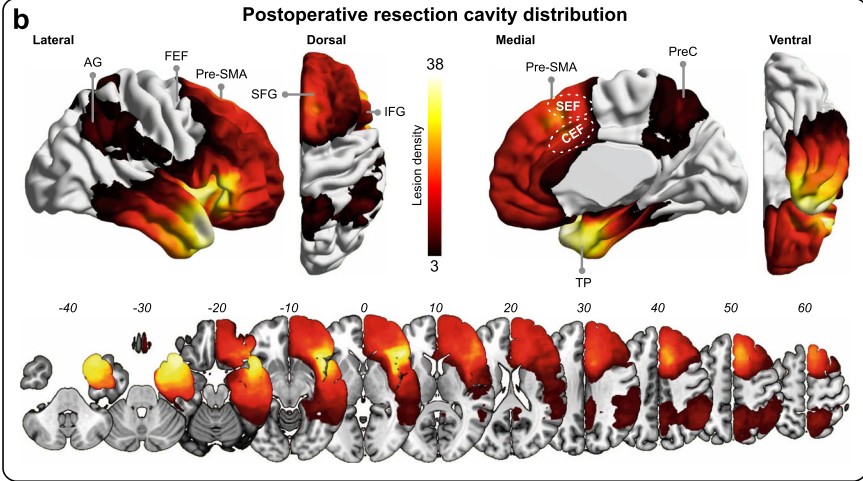

**Fig. 1 Lesion overlap maps.** Density overlap map of **a** preoperative tumors and **b** surgical cavities for the 128 patients included in this study. These maps are thresholded in such a way that only voxels affected in at least three patients (the threshold for SVR-LSM analyses; see "Methods" section) are shown. Bars indicate lesion density. The maximum overlap was in the insula for both the tumor map ($n = 52$) and the resection cavity map ($n = 38$). The relative position of the supplementary eye field (SEF) and the cingulate eye field (CEF) is shown. For indicative purposes, both areas were more or less infiltrated in 16 patients before surgery and furthermore or less resected in 18 patients. AG angular gyrus, FEF frontal eye field, IFG inferior frontal gyrus, PreC precuneus, Pre-SMA pre-supplementary motor area, SFG superior frontal gyrus, TP temporal pole.

comparing Δ1 (mean $1.91 \pm 5.38$) and Δ2 (mean $0.18 \pm 3.01$) ($t_{(127)} = 4.34$, $p < 0.001$; two-sided; 95% CI [0.95, 2.53]) (Fig. 2b).

The same analyses were repeated, first with total_bell. RM ANOVA confirmed that the total number of omitted bells significantly differed across measures ($F_{(2, 254)} = 45.05$, $p < 0.001$, $\eta^2_p = 0.26$; two-sided) (Fig. 2c). It was greater just after surgery (A1 vs A2; mean $2.19 \pm 2.61$ vs $5.59 \pm 5.24$; $p < 0.001$; two-sided; 95% CI [$-4.44$, $-2.35$]), but not 3 months after (A1 vs A3; mean $2.19 \pm 2.61$ vs $2.04 \pm 2.33$; $p = 0.93$; two-sided; 95% CI [$-0.89$, $-1.19$]) – meaning that patients fully recovered in average. The difference between Δ1 (mean $3.39 \pm 5.84$) and Δ2 (mean $-0.16 \pm 2.94$) was strongly significant ($t_{(127)} = 7.89$, $p < 0.001$; two-sided; 95% CI [2.66, 4.45]) (Fig. 2d).

Last, the asymmetry score (i.e., diff_bell) also evolved across the three assessments ($F_{(2, 254)} = 30.34$, $p < 0.001$, $\eta^2_p = 0.19$; two-sided). Compared to the preoperative baseline, it was greater immediately after surgery (A1 vs A2; mean $0.04 \pm 1.48$ vs $2.35 \pm 4.17$; $p < 0.001$; two-sided; 95% CI [$-3.13$, $-1.49$]) but comparable 3 months later (A1 vs A3; mean $0.04 \pm 1.48$ vs $0.22 \pm 1.52$; $p = 0.86$; two-sided; 95% CI [$-0.99$, $-0.63$]) (Fig. 2e). Accordingly, the difference between Δ1 (mean $2.31 \pm 4.36$) and Δ2 (mean $0.18 \pm 2.07$) was significant ($t_{(127)} = 5.60$, $p < 0.001$; two-sided; 95% CI [1.36, 2.91]) (Fig. 2f).

In summary, neurosurgeries impaired task performance, but only in the immediate postoperative period. For the bell test, items situated on the left side were considerably more affected than those placed on the right side (i.e., a significant increase of diff_bell) – a typical sign of spatial neglect.

To determine whether patterns of performances converged between both tasks, simple non-parametric correlations were performed. Only a slight association was observed between line bisection estimates and diff_bell at A1 ($r_{128} = 0.23$, $p = 0.008$; two-sided) and for Δ1 ($r_{128} = 0.25$, $p < 0.001$; two-sided), suggesting that the underlying impaired neurocognitive mechanisms did not fully overlap across the two tasks. Note that this correlation disappeared when Δ2 was considered ($r_{128} = 0.14$, $p = 0.11$; two-sided). Correlation matrices are displayed in Supplementary Table 6.

**SVR-LSM results**. A grid searching approach was used to determine the optimal hyper-parameters (i.e., $\gamma$ and $C$) of SVR-LSM models (see "Methods" section). With respect to A1, we failed to identify a combination of hyper-parameters associated with a good prediction accuracy and a high level of reproducibility for the three measures of interest (see Supplementary Fig. 3a for grid search). This was related to the fact that there was only a

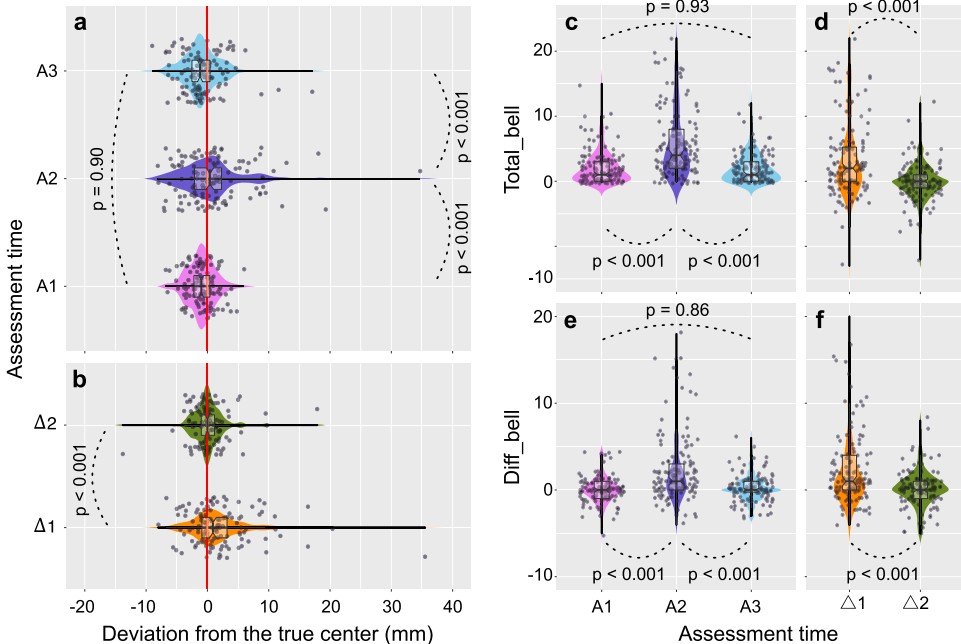

**Fig. 2 Combined violin and notch boxplots of behavioral measurements. a** The three assessments for the line bisection task ($n = 128$ for each measure). **b** Delta measures for the line bisection task ($n = 128$ for each measure). **c** The three assessments for *total_bell* ($n = 128$ for each measure). **d** Delta measures for *total_bell* ($n = 128$ for each measure). **e** The three assessments for *diff_bell* ($n = 128$ for each measure). **f** Delta measures for *diff_bell* ($n = 128$ for each measure). For each notch boxplot, the horizontal line at the center indicates the median value, the lower and upper bounds of the box indicate 25–75% interquartile range and whiskers indicate 1.5× interquartile range; individual scores are shown separately as dots. Repeated measures ANOVAs (two-sided) were used to assess statistical significance. When significant, pairwise multiple comparisons analyses were conducted with the Scheffé test which controls for the familywise error rate. Statistical analyses are fully detailed in the main text. R software (https://www.R-project.org/; packages = *ggplot2* & *ggpubr*) was used to create this figure. Source data are provided as a Source Data file.

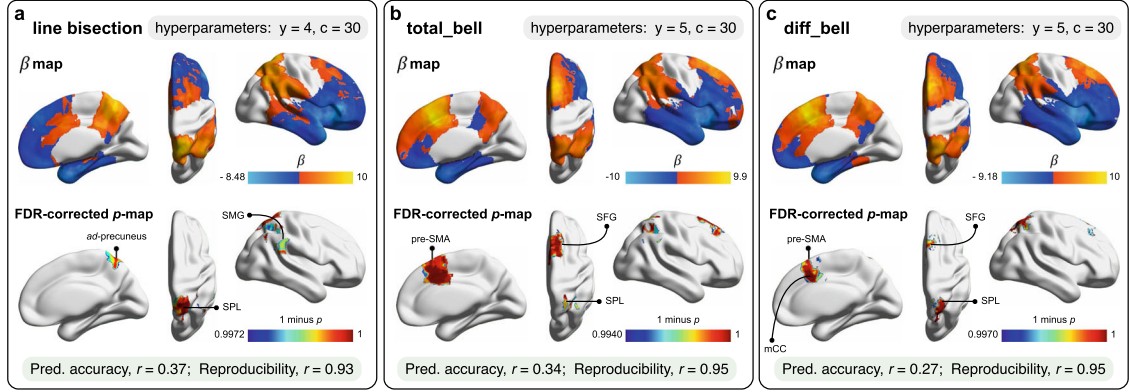

**Fig. 3 SVR-LSM results. a** line bisection estimates. **b** total_bell. **c** diff_bell the raw β-map is placed at the top, whereas the FDR-corrected (at $q = 0.05$) (1-p)-map is underneath. The used hyper-parameters are displayed at the top. The prediction accuracy and reproducibility of the SVR-LSM models are indicated at the bottom. ad antero-dorsal, GSM supramarginal gyrus, pre-SMA pre-supplementary motor area, SFG superior frontal gyrus, SPL superior parietal lobule. Note that 1-*p*-values are used to facilitate visualization. See Table 1 for a detailed report of significant areas.

limited amount of pathological variance to be modeled given the little or the absence of behavioral differences between the control and the patient group at preoperative baseline (see Supplementary Figs. 1 and 2)—in agreement with the established brain's efficient abilities to reorganize in response to lower-grade glioma progression[40]. By contrast, satisfactory hyper-parameters were identified for Δ1 for all measurements of visuo-spatial attention, including line bisection estimates ($\gamma = 4$, $C = 30$) total_bell ($\gamma = 5$, $C = 30$) and diff_bell ($\gamma = 5$, $C = 30$) (see Supplementary Fig. 4 for grid search). The goodness-of-fit and reproducibility of selected models are displayed in Fig. 3. The generated SVR-LSM maps located clusters of suprathresholded voxels for both tasks, meaning that removal of specific brain structures had specific and

immediate consequences on visuo-spatial attention. With regard to the bisection line task, rightward deviations were uniquely associated with areas of the parietal cortex (see Fig. 3a for both the raw β-map and the subsequent corrected statistical map). The most significant and large brain-behavior relationship was observed in the superior parietal lobule (SPL), extending to the inferior parietal cortex (including both the supramarginal and angular gyri), and medially to the precuneus (see Table 1 for a detailed report of the results). The SVR-LSM model for total_bell revealed a dissociated pattern of associations (Fig. 3b): while areas of the parietal cortex were still identified, including the inferior and superior parietal lobules, the most significant cluster of suprathresholded voxels was detected on the medial face of the

**Table 1 SVR-LSM results for Δ1.**

| AAL parcels | Significant voxels (n) in the parcel | Parcel percentage with significant voxels | Average p-values for significant voxels |
|---|---|---|---|
| _Line bisection_ estimates (rightward deviations): $p_{FDR(q=0.05)} < 0.0028$ | | | |
| Parietal_Sup | 7979 | 45.5% | 0.00051 |
| Precuneus | 3532 | 13.5% | 0.00088 |
| Angular | 2933 | 20.9% | 0.00103 |
| Parietal_Inf | 2325 | 21.6% | 0.00124 |
| SupraMarginal | 3523 | 22.3% | 0.00149 |
| _total_bell_ (total number of omitted bells): $p_{FDR(q=0.05)} < 0.004$ | | | |
| Cingulum_Mid | 5844 | 33.5% | 0.00043 |
| Supp_Motor_Area | 7709 | 40.8% | 0.00096 |
| Frontal_Sup | 7227 | 22.5% | 0.00103 |
| Frontal_Sup_Medial | 2124 | 12.5% | 0.00162 |
| Parietal_Inf | 1731 | 16.1% | 0.00217 |
| Parietal_Sup | 2611 | 14.9% | 0.00227 |
| Angular | 2417 | 17.3% | 0.00310 |
| _diff_bell_ (left minus right bells): $p_{FDR(q=0.05)} < 0.003$ | | | |
| Parietal_Sup | 4999 | 28.5% | 0.00038 |
| Parietal_Inf | 874 | 8.1% | 0.00061 |
| Cingulum_Mid | 2980 | 17.1% | 0.00063 |
| Supp_Motor_Area | 2881 | 15.3% | 0.00105 |
| Frontal_Sup_Medial | 1054 | 6.20% | 0.00121 |
| Frontal_Sup | 1707 | 5.3% | 0.00127 |

The permutation-derived p-maps were thresholded with a FDR procedure (see "Methods" section). Only areas harboring suprathresholded voxels in a proportion of at least 5% are detailed.

frontal lobe, including the supplementary motor area (SMA) and the middle cingulate cortex, extending to both the medial and the superior frontal gyri (Table 1). Importantly, the same pattern of results was also observed when diff_bell was considered, with however a higher degree of anatomo-functional specificity (Fig. 3c and Table 1).

In accordance with previous behavioral analyses indicating that patients regained in average their preoperative baseline performances (see above), no combinations of hyper-parameters were found to be associated with good model fit and reproducibility for Δ2 (Supplementary Fig. 3b).

**Group analyses**. To better highlight the dissociation described above, we directly contrasted the behavioral performances of all patients with a resection located in the parietal cortex ($n = 18$) versus located in the medial frontal lobe ($n = 21$) (Fig. 4a). A two-way mixed ANOVA performed on the line bisection performance showed, as expected, a principal effect of group ($F_{(1, 37)} = 5.92$, $p = 0.02$, $\eta^2_p = 0.14$; two-sided) and assessment time ($F_{(2, 74)} = 24.57$, $p < 0.001$, $\eta^2_p = 0.40$; two-sided), and most importantly a significant interaction effect between both factors ($F_{(2, 74)} = 12.86$, $p < 0.001$, $\eta^2_p = 0.26$; two-sided). Post-hoc analyses revealed that performances differed between both groups, but only at A2 ($p < 0.001$; two-sided; 95% CI [1.17, 14.15]; $p > 0.10$ for comparisons at A1 and A3) (Fig. 4b). With respect to diff_bell, a principal effect was found for assessment time ($F_{(2, 74)} = 29.05$, $p < 0.001$, $\eta^2_p = 0.44$; two-sided) but not for group ($F_{(1, 37)} = 0.56$, $p = 0.46$, $\eta^2_p = 0.015$; two-sided). Both factors did not interact significantly ($F_{(2, 74)} = 0.72$, $p = 0.49$, $\eta^2_p = 0.019$; two-sided) (Fig. 4c). The same pattern of results was found for total_bell (Supplementary Fig. 5).

In summary, the above analyses confirmed that the bell test was affected to the same extent by resections targeting either the parietal or the fronto-medial areas. By contrast, line bisection performances were uniquely impaired following parietal resections. This general finding is reflected in the individual patterns of performances (Supplementary Table 7).

**Tract-level analyses**. To ascertain whether surgically-related damage of specific white-matter tracts accounted at least partially

for the attention bias we described above, measures of disconnection severity (i.e., the number of streamlines interrupted by the surgical procedure) were computed for each candidate tract (see "Methods" for the procedure of selection) and correlated with behavioral measurements, in particular, Δ1 and Δ2.

The results are illustrated in the form of a correlogram in Fig. 5. Here we interpret only positive correlations for which the critical p-value (two-sided) was reached after Bonferroni correction (i.e., $p = 0.002$). The amount of resected fibers within the superior thalamic radiations ($r_{128} = 0.29$, $p < 0.001$), SLF_I ($r_{128} = 0.29$, $p < 0.001$), fronto-parietal cingulum ($r_{128} = 0.28$, $p = 0.0014$), and within SLF_II ($r_{128} = 0.27$, $p = 0.0022$), positively correlated with Δ1 for the line bisection task. No significant correlations were found for Δ2.

With regard to diff_bell, only two tracts were found to be associated with Δ1, including again, albeit more markedly, SLF_I ($r_{128} = 0.35$, $p < 0.001$) and fronto-parietal cingulum ($r_{128} = 0.31$, $p < 0.001$). No significant correlations were found for Δ2.

The analyses were repeated for total_bell; a larger set of tracts was associated with the behavioral measures, especially for Δ1. This included SLF_I ($r_{128} = 0.42$, $p < 0.001$), superior thalamic radiations ($r_{128} = 0.42$, $p < 0.001$), fronto-parietal cingulum ($r_{128} = 0.39$, $p < 0.001$), superior CST ($r_{128} = 0.31$, $p < 0.001$) and, marginally, fronto-para-hippocampal cingulum ($r_{128} = 0.27$, $p = 0.0025$). For Δ2, only fronto-para-hippocampal cingulum ($r_{128} = 0.32$, $p < 0.001$) and marginally SLF_I ($r_{128} = 0.275$, $p = 0.0029$) still correlated.

**Case study**. Here, we separately analyze the case of a patient in whom two-stage wide-awake neurosurgery was performed to remove a lower-grade glioma mainly infiltrating the right cingulum bundle (see Supplementary Fig. 6 for the patient's native MRI). To access the anterior part of the tumor, a first surgery was achieved through a trans-SFG approach. Both the pre-SMA (medially) and the posterior part of SFG (laterally) were resected (Fig. 6a). In line with the results from SVR_LSM, the patient experienced spatial neglect, but only transiently. Most importantly, only the bell test was affected, almost exclusively for bells located on the left side (Fig. 6b). During the second surgery

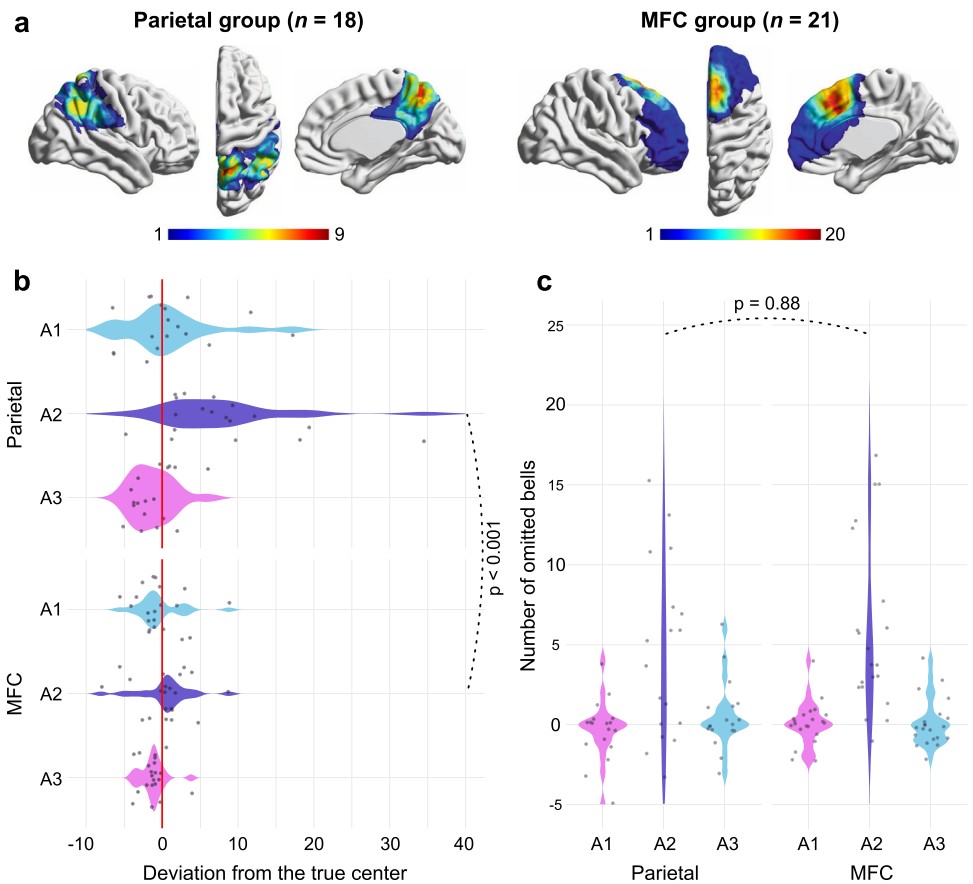

**Fig. 4 Results from group analyses. a** Overlap maps for each group or patients. Bars indicate lesion density at each voxel. **b** Violin plot for line bisection performance. **c** Violin plot for diff_bell. Individual scores are shown separately as dots. Two mixed ANOVAs (two-sided) were conducted to assess statistical significance, with the time of assessment {A1, A2, A3} as a within-subject factor and patient group as a between-group factor {parietal, MFC}. When significant, pairwise multiple comparisons analyses were conducted with the Scheffé test which controls for the familywise error rate. Statistical analyses are fully detailed in the main text. Source data are provided as a Source Data file.

performed 6 months later, removal of the anterior part of both the precuneus and the SPL was accomplished to access the posterior part of the tumor (Fig. 6a). Performances in both tasks were severely impacted immediately after surgery and remained impaired 3 months after (Fig. 6b). Taken as a whole, these results confirmed the dissociation highlighted above.

Tract-level analyses (Fig. 6c) confirmed that the first surgery caused a strong disconnective breakdown of the medial tracts, including the SLF_I, the different layers forming the cingulum and, to a much less extent, the FAT, the mid-anterior part of the corpus callosum, the superior cortico-striatal tract, the fronto-pontine tract, and the superior thalamocortical radiations. During the second surgery, both the middle-to-posterior and posterior parts of the corpus callosum, as well as the extreme capsule, were mainly interrupted. The following tracts were less severely damaged: middle longitudinal fasciculus, parieto-pontine tract, medial lemniscus, posterior thalamic radiations, and posterior cortico-striatal tract. Interestingly, the lasting spatial neglect observed following the second surgery could be hardly explained by a disruption of the SLF_1 or the cingulum, as these tracts were likely to be "disconnected" during the first surgery.

## Discussion

Lesion mapping works remain an essential means of ascertaining how critical is a brain structure within a given functional network[53,54]. Admittedly, however, the findings that emerge from these studies are highly dependent on the lesion model upon

which lesion-deficit inferences are drawn. Because stroke is a common condition, most of the available neuropsychological literature is based on the performances of patients suffering from this acute lesion that mainly distributes around the perisylvian sulcus[36]. The main consequence is that the functional contribution of specific cerebral structures with poor or no lesion coverage is underestimated or not discussed at all. In the domain of visuospatial cognition, this inherent shortcoming has caused difficulties in gauging the implication of DAN-related structures in spatial neglect[37], a well-studied but still ill-defined syndrome in terms of anatomical correlates[43]. In the present study, we capitalized on a large and longitudinal neuropsychological dataset gained from patients with a slow-growing tumor to test the hypothesis whereby structures shaping the right medial network may play an important role in the voluntary deployment of visuospatial attention. Overall, our results substantiated this contention, as surgical removal of structures in a location compatible with the SEF and CEF was specifically associated with difficulties in purposely exploring the contralesional space in a visual search task.

As already suggested by Mesulam thirty years ago[1], spatial neglect is better conceptualized as a multicomponent syndrome whose anatomical correlates may differ as a function of the used tasks that generally differ in terms of cognitive requirement[55]. This view is not only supported by the various dissociations that have been reportedly observed between task performances but also by the failure to identify a critical anatomical locus for spatial neglect—despite the wealth of available neuropsychological

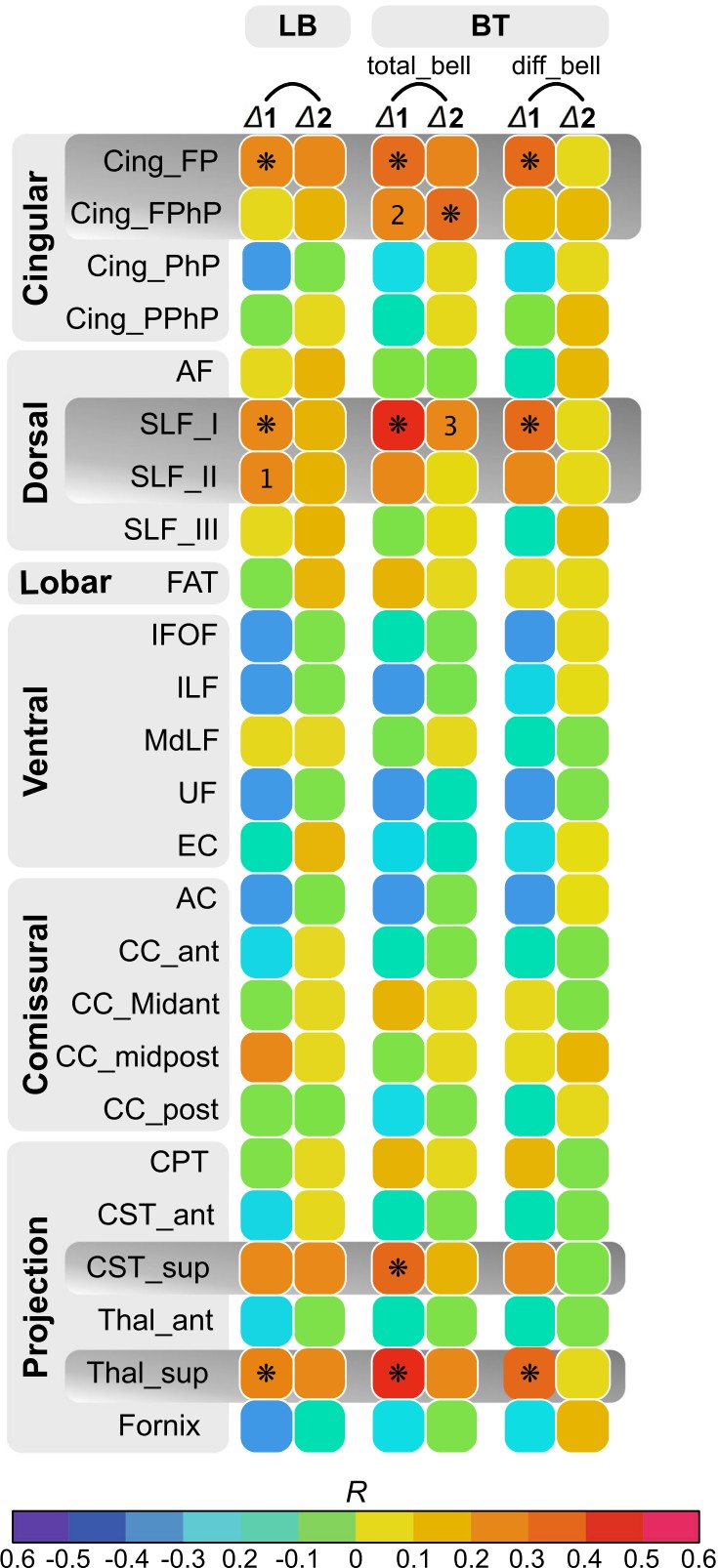

**Fig. 5 Correlograms showing the relationships between the behavioral measures and the amount of damage of white-matter tracts.** * Means significant positive correlations after Bonferroni correction (critical *p*-value = 0.002). FP fronto-parietal; FPhP fronto-parahippocampal; PhP parahippocampal, AF arcuate fasciculus, SLF superior longitudinal fasciculus, FAT frontal aslant tract, IFOF inferior fronto-occipital fasciculus, ILF inferior longitudinal fasciculus, MdMF middle longitudinal fasciculus, UF uncinate fasciculus, EC extreme capsule, AC anterior commissure, CC corps callous, CPT cortico-pontine (frontal), CST Cortico-striatal tract, Thal thalamus, LB line bisection, BT bell test. 1, *p* = 0.0022; 2, *p* = 0.0024; 3, *p* = 0.003. Source data are provided as a Source Data file.

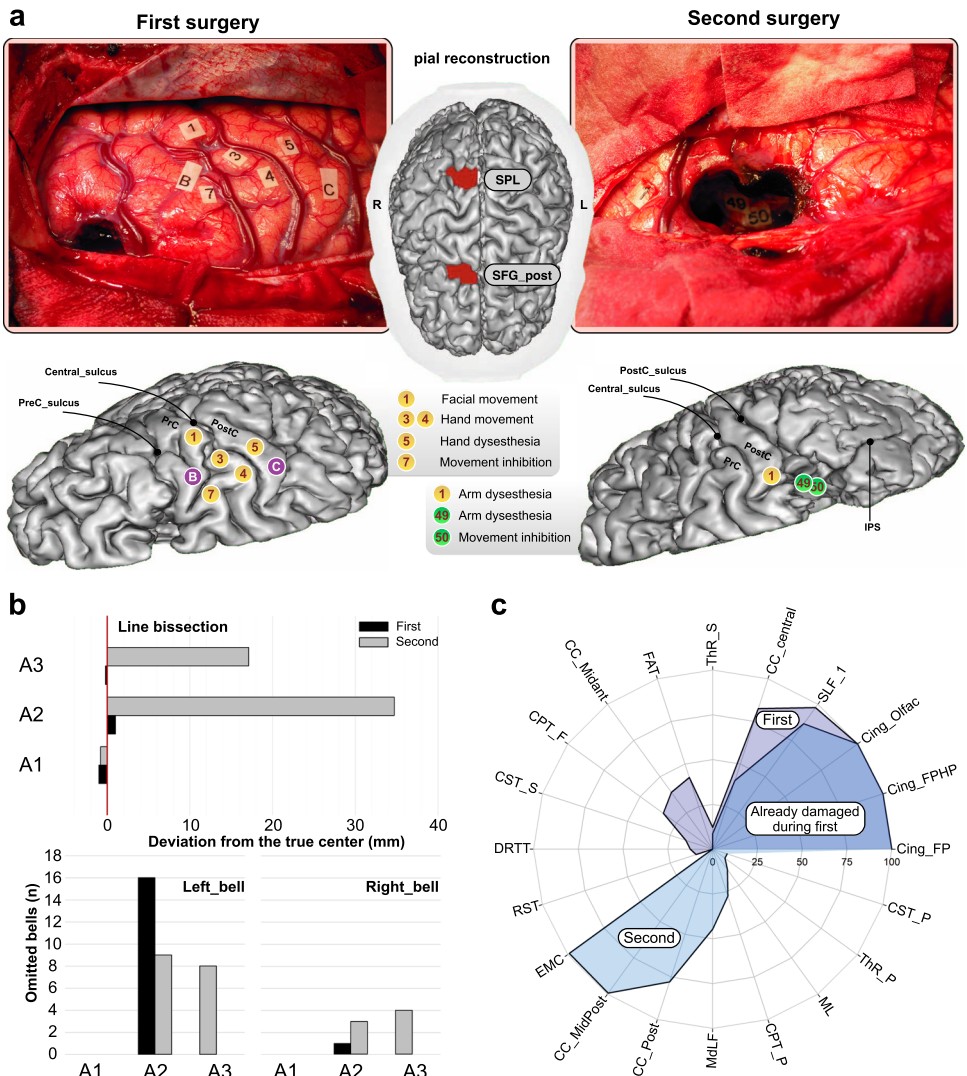

**Fig. 6 Anatomical, neurosurgical, and behavioral data of the patient analyzed in isolation (case study). a** Intrasurgical photos and pial reconstruction of the brain (performed on the 3-month postoperative MRI). Labeled numbers correspond to functional sites unmasked during electrostimulation mapping, while labeled letters correspond to spatial delineation of the tumor obtained with ultrasonography. The red color on the central pial mesh indicates the spatial location of the two successive resections. **b** Behavioral data at each time point and for each surgery. **c** Radar plot indicating the estimated percentage of damage within each tract for each surgery. CC corpus callous, Cing cingulum, CPT cortico-pontine tract, CST cortico-striatal tract, DRTT dentatorubrothalamic tract, EMC extreme capsule, FAT frontal asltant tract, MdLF middle longitudinal fasciculus, ML medial lemniscus, PrC precentral gyrus, PostC postcentral gyrus, RST reticulospinal tract, SFG superior frontal gyrus, SLF superior longitudinal fasciculus, SPL superior parietal lobule, ThR thalamic radiations. See Supplementary Fig. 6 for slices from the patient' native anatomical MRI. Source data are provided as a Source Data file.

findings[43]. One of the issued hypotheses that has been gaining attraction over the years, but not fully evidenced by experimental data, is that perceptual tasks (such as the line bisection task) may be preferentially affected by lesions damaging the posterior parietal cortex, whereas visuo-motor exploratory tasks (such as the bell test) may rather be affected by lesions targeting the frontal lobe[1,43,44,46]—knowing that this general pattern may be in reality more complex as cancellation performances have been also found to be impaired following posterior or subcortical lesions (e.g., refs. [56,57]). Our results contribute to this old but still topical and clinically relevant matter by indicating that such a simple fronto-parietal dissociation is indeed just part of the story. While rightward biases in line bisection were indeed associated with a cluster of parietal areas lodging in both the inferior and superior parietal lobules, bell omissions (total_bell and diff_bell) were associated with both the posterior parietal cortex (albeit with a lesser extent) and the fronto-medial cortex (in particular the pre-

SMA and the anterior-to-middle cingulate). This finding was unequivocally replicated in the patient who benefited from a two-step surgery and further supported by the low-to-mild correlations found between task performances at the population-level.

The critical finding of the current study is the powerful and to date undocumented association between the medial frontal cortex and the emergence of visuo-motor exploratory neglect. From an anatomical standpoint, the cluster of significant voxels identified by SVR-LSM greatly overlapped with the consensual location of both the SEF (junction between the SMA and the pre-SMA) and the CEF (the most caudal/rostral part of the anterior/middle cingulate). To date, the respective role of these two eye-related medial areas remains controversial, and almost undiscussed in the context of visuo-spatial attention. Studies performed in humans and monkeys converge towards the idea that the SEF may significantly contribute, not to saccade initiation per se[58], but to eye behavior monitoring and conflict signaling. Likewise, the CEF

may be indirectly involved in the control and regulation of saccadic eyes movements by motivational influence[9], stemmed from the consequences of previously accomplished voluntary eye movements[59]. The role of the CEF in motivation may extend to visuo-spatial attention, but its lesion in isolation is apparently not enough to produce contralesional motivational neglect[60]. Our findings clearly demonstrate that damage to these two eye-related areas is able to produce visuo-motor exploratory neglect, though the underlying mechanisms cannot be fully elucidated here; it might be either the consequence of a deficit in initiating saccades towards the contralesional space, in line with the established behavioral affiliations of the medial frontal structures[61], or the result of specific difficulties in deliberately orienting attention during visual search (i.e., the attentional function attributed to the FEF). The latter interpretation is more likely since ablation or inactivation studies in monkeys have generally shown not or only little impact of SEF/CEF damage on saccade initiation[58,62]. Finally, we cannot fully exclude the possibility that visuo-motor exploratory neglect partially arises from a functional diaschisis effect[63] transiently impairing the FEF by deprivation of functional inputs—the FEF, SEF, and CEF being densely interconnected[21,22].

At the subcortical level, no unequivocal task-specific disconnectivity patterns were captured by the analyses, but dissimilarities were nevertheless observed. First, although disruptions of thalamic radiations, SLF_I, and fronto-parietal cingulum all contributed to impaired performances in both tasks, the correlations observed for the two latter tracts were greater for total_bell/diff_bell vs. line bisection—in line with our prior hypotheses. Second, line bisection performances were also affected by SLF_II, a result that has been already highlighted in neurostimulation[28,32] and lesion mapping[27] studies. Third, target cancellation performances (only total_bell) were associated with disconnective interruption of the superior cortico-striatal tract. This result is interesting to consider in view of its connective pattern. This tract indeed provides dense connections between the fronto-medial cortex (in particular, SMA and pre-SMA, perhaps the FEF) and the caudate, and is known to be part of the eye-related network[64]. Its inactivation through electrostimulation leads to contralateral eye deviation[16]. Moreover, the caudate influences visually-guided actions[65] and its damage causes spatial neglect in humans[23]. To summarize, disruption to the medial fiber network and the cortico-striatal connections are more likely to disturb target visual search. This conclusion is bolstered by the observation that the likely interruption of the fibers forming the SLF_I and the cingulum following the first surgery did not result in rightward deviations in the single patient (Supplementary Notes 1 and 2).

The powerful but transitory effects we observed may be fairly interpreted as evidence that the structures in question are not crucial for visuo-spatial attention. A number of explanations may be put forward, these ones not being mutually exclusive. First and foremost, the slow-growth kinetics of lower-grade glioma is known to favor efficient neuroplastic compensation which takes place upstream, before neurosurgeries are performed[66]. This explains, on the one hand, that pathological variance was not enough to be reliably modeled by SVR-LSM preoperatively and suggests on the other hand, that removing structures that have been progressively overwhelmed may circumscribe the behavior effects of surgeries in a narrowed window of time—because the attention network has already but not entirely reorganized. For example, it is a common clinical observation that surgical excision of the SMA causes akinetic mutism, but only transiently. This fast recovery is associated with a "reallocation" of the SMA in the contralesional hemisphere, suggesting that the brain is capable of instantiating dynamic strategies of homotopic compensation[67]. Second, and in connection with this, structures forming the DAN may be innately more robust to damage because it is bilaterally distributed contrary to the VAN[51]. This line of explanation is supported by reports showing that unilateral ablation or pharmacological inactivation of the FEF or the SEF in rhesus monkeys have only transient effects on visuo-spatial attention or oculomotor behaviors[62]. Third, the patient sample on which we relied is not unbiased as, for some of them, an intraoperative mapping of visuo-spatial attention was performed in an attempt to safely remove tumors lodged in the posterior parietal cortex. This may explain for example why we only found a limited and transitory effect for SLF_II which is typically mapped and spared to avoid lasting postoperative neglect[28,32]. However, this does not apply to regions outside the parietal cortex for which visuo-spatial attention was not intraoperatively monitored.

In conclusion, our findings provide evidence for a significant role of the medial eye field network in the voluntary deployment of attention, as its disruption causes unilateral visuo-motor exploratory neglect. These findings have important implications for current neurocognitive and neurocomputational models of visuo-spatial attention.

## Methods

**Standard protocol approvals and participant consents.** All patients (including the case study described below) were evaluated in the context of their standard medical care, and gave their informed consent to participate in this study. The study protocol was approved by Montpellier University Medical Center' institutional review board (N°202000557). All control participants who were enrolled retrospectively also provided informed consent to participate. IRB approval for this part of the study was obtained from the French College of Neurosurgery (N°00011687).

**Patient sample.** The sample consisted of 128 patients consecutively operated on for a lower-grade glioma invading the right hemisphere at University Montpellier Medical Center' Department of Neurosurgery over a period of 7 years (2013–2020). Patients fulfilling the following exclusion criteria were discarded at the outset: higher-grade glioma identified by histopathological analyses, adjuvant radiotherapy performed before or after surgery, a visual hemianopsia identified before or after surgery to avoid contaminating task performance, and a lack of longitudinal behavioral data. The patients' sociodemographic and clinical characteristics are given in Supplementary Table 1. Note that none of the patients suffered from visual extinction before, 4 days after, and 3 months after surgery. It was assessed using a double simultaneous stimulation test (i.e., visual stimulations were performed with a finger either unilaterally, left or right, or bilaterally).

A control group of 44 neurologically healthy participants matched in terms of age, educational attainment, and sex was further recruited to assess differences with the patient group (Supplementary Table 4).

**Surgical procedure.** In agreement with our standard surgical approach, all patients were operated on in awake condition with a multifunctional intraoperative mapping performed by means of direct electrostimulation[68]. This surgical technique has been thoroughly described elsewhere, and is beyond the scope of this study that did not include stimulation data. However, it is worth mentioning that all patients with a parietal tumor except one benefited from an intraoperative mapping of visuo-spatial attention conducted with a line bisection task to avoid long-lasting, postoperative spatial neglect[28]. This issue has been taken in full consideration in the interpretation of the data. Note that the only patient who did not benefit from this mapping is the one described in the section "case study".

**Behavioral tasks and measures.** Patients performed the behavioral assessment at three time points: the day before surgery (noted elsewhere A1 for assessment one), 4 days after surgery (noted elsewhere A2) and 3 months after surgery (noted elsewhere A3). They were asked to complete two well-tried visuo-spatial tasks, the performances of which are known to be affected by spatial neglect: the line bisection task[69] and the bell test[70]—the latter being a target cancellation task. With respect to the former, patients were required to estimate the true midpoint of ten horizontal lines of 18 cm in length and 3 mm in thickness. The task was adapted in a touchscreen environment so that patients used a digital stylus with the dominant hand to mark the estimated center of lines. Deviations from the true center were automatically computed across trials and then averaged to form the final measure. Patients with left spatial neglect behave in such way that the estimated center significantly shifts toward the right. In the bell cancellation task, patients are asked to encircle 35 bells among 280 distractors on a sheet of A4 paper displayed in landscape format. 17 bells are homogenously distributed on both sides, and one bell is placed in central position. In this study, there was not limit of time to perform the task. Patients were simply asked to notify the experimenter when s/he thought s/he had circled all of the bells. The total number of omitted bells, as well

as the asymmetry score (i.e., the difference between the number of omitted bells on the left side and the number of omitted bells on the right side), were taken into consideration as two separate dependent variables. Patients with spatial neglect typically omit the bells situated on the left side, with some degree of variability as a function of neglect severity.

As we had in this study, a baseline assessment before the surgery was performed, most of the analyses described in the following were achieved on the preoperative performances (*effect of tumor invasion*), the delta between the preoperative and the 4-day postoperative performances (noted elsewhere *Δ1*; *early effect of surgical resection*), and the delta between the preoperative and the 3-month postoperative performances (noted elsewhere *Δ2*; *late effect of surgical resection*). In this way, we had the chance to more directly appraise the consequences of surgical excisions on spatial attention.

**Neuroanatomical data and lesion drawing**. In this study, two MRI sequences were used to map the neuroanatomical data. In particular, FLAIR images were used to map the preoperative tumors because this sequence is known to yield the best contrast between normal *vs.* infiltrated brain tissue. Conversely, high-resolution whole-brain $3DT_1$ images acquired at 3-month postsurgery (in the context of the standard care) were preferred to map the surgical cavities. The imaging parameters were as follows: (i) preoperative FLAIR images (1.5 T/3 T): repetition time, 13,200/800 ms; echo time, 109/108 ms; inversion time, 2500/23,700 ms; field of view 210 × 240/202 × 240 mm, voxel size 0.898 × 0.898 × 6 mm$^3$, slice thickness 5/3 mm, spacing 5.5/3.6 mm, and flip angle 150°; (ii) 3-month 3DT1 images (1.5 T/3 T): repetition time, 1880/1700 ms; echo time, 3.4/2.5 ms; inversion time, 1100/922 ms; field of view, 256 × 256 mm; voxel size, 1 × 1 × 1 mm$^3$, 176 axial slices, and flip angle 15°/9°. Note that the use of two different magnets was independent of the general purpose of this study.

To minimize the potential bias caused by abnormal lesion-related radiological signals, MRI datasets were registered to the MNI space using enantiomorphic normalization[71]. This procedure was performed with the SPM12 (https://www.fil.ion.ucl.ac.uk/spm/software/spm12/) Clinical Toolbox (https://github.com/neurolabusc/Clinical)[72]. The output resolution was 1*1*4 mm for FLAIR images and 1-mm isometric for 3DT1 images. As a first step, the tumors/resection cavities were semi-automatically drawn using MRIcron package (https://github.com/neurolabusc/MRIcron) and further inflated by means of a three-dimensional smoothing procedure (3-mm full-width-at half-maximum [FWHM] Gaussian kernel with a threshold of 0.3). The obtained masks were then binarized and inserted during the registration process (during enantiomorphic normalization, the area covered by a particular mask is replaced by the undamaged homologous area within the contralesional hemisphere). Before proceeding further, all normalized MRIs were systematically and carefully checked to identify and potentially exclude inaccurate registrations. All were satisfactory at this stage. Next, tumors and resections cavities were drawn again on the normalized MRIs, yielding two three-dimensional volumes of interest (VOI) by patients. These VOIs were spatially smoothed with a 2-mm FWHM Gaussian kernel (threshold of 0.4). The whole procedure was performed by the same experimenter who shows highly-skilled expertise in neuro-anatomy (the first author).

**Multivariate lesion-symptom mapping**. Support vector regression-based lesion-symptom mapping (SVR-LSM)[47,48] was used to explore the relationship between the location of tumors or resection cavities and behavioral measures of visuo-spatial attention. Contrary to standard mass-univariate voxel-based lesion-symptom mapping (VLSM) that assumes statistical independence across voxels[49], SVR-LSM rather works at the level of the entire lesion map thus taking into consideration the necessary inter-dependent nature of the lesioned voxels. In practice, a non-linear function is used to train a SVR model, the goal of which is to predict as precisely as possible the behavioral scores using all voxels' lesion statuses simultaneously. This multivariate lesion mapping method may reach better sensibility and specificity over standard univariate approach[47], especially when the sample size is large enough[50], and is associated with a lower rate of false-positive outcomes[73]. It appears however that standard but conservative univariate LSM may remain a practical option if conducted with large populations[73].

In this study, we used the Matlab script originally coded by Zhang et al.[47] to perform all SVR-LSM analyses (https://github.com/yongsheng-zhang/SVR-LSM). *Epsilon*-SVR models with a radial basis kernel function (RBF) were used to estimate hyperplane. Hyper-parameters of SVR-LSM models were optimized using a grid searching procedure, in particular the cost ($C$) which corresponds to the penalty/regularization parameter and gamma ($γ$) which represents the kernel coefficient. This optimization was performed by means of a 5-fold cross-validation procedure allowing to determine the combination of parameters that maximized prediction accuracy while maintaining a high level of reproducibility (see Zhang et al.[47] for a complete description of how model fit and reproducibility are estimated). In total, 66 couples of parameters were assessed for each behavioral measure of interest and for each time point i.e., *A1*, *Δ1* and *Δ2* (9 optimization procedures, sum-total), with $C = [1\ 10\ 20\ 30\ 45\ 50]$ and $γ = [0.1\ 1\ 2\ 3\ 4\ 5\ 6\ 7\ 8\ 9\ 10]$. Parameter assessment was done with a publicity available Matlab script ('svr_lsm_BasicScript_opt.m', https://data.mendeley.com/datasets/2hyhk44zrj/2)[74] after it was checked for quality and modified for local use. Datasets associated with a poor prediction accuracy

($r_{max} < 0.20$) and/or an insufficient index of reproducibility ($r_{max} < 0.90$) were not eligible to subsequent SVR-LSM analyses.

Prior to running SVR-LSM analyses, the DTLVC (direct total volume control) option was chosen to control for lesion volume directly to the lesion data (and not to the behavior performances). Note that this parameter was also taken into consideration during the optimization procedure of hyper-parameters. A bootstrap permutation procedure (5000 permutations) was first used to estimate the significance of the voxels' feature weight. Then, a FDR correction[75] ($q = 0.05$) (directly implemented in the script by Zhang et al.[47]) was applied to threshold the resulting SVR-LSM *p*-maps. An extent threshold of 50 voxels was employed. Note that voxels were analyzed if they were lesioned in at least three patients. This rather low cut-off was originally selected to include structures relevant to spatial cognition, but typically less affected in the context of lower-grade glioma (i.e., some regions of the parietal cortex[40]). Results are reported within the AAL atlas[76].

**Tract-level disconnection analyses**. To estimate the severity with which the main white-matter tracts were damaged by the surgical resection, we used the lesion quantification toolbox (LQT)[77]—a new Matlab toolbox that employs different population-based approaches to provide multiple measures of white-matter disconnection severity, including tract-level disconnection measures (Supplementary Note 3). The resection cavities maps were embedded into the HCP-1065 tractography atlas (http://brain.labsolver.org/diffusion-mri-templates/tractography; see ref. [78] for atlas construction; atlas version April 2020) and used as regions-of-interest. The used atlas has the clear advantage of being constructed on an unprecedented sample of subjects. Moreover, compared to older versions (i.e., HCP-842), it integrates clear distinctions between the different strata of the superior longitudinal fasciculus (which is central for our study) and between the different pathways forming both the striatal and thalamic projection systems. As detailed in Griffis et al.[77], the fibers of each HCP tract are loaded and filtered in such a way that only fibers intersecting the lesion map are retained. For each tract, an estimate of disconnection severity (in percentage) is provided. Compared to other methods based on the amount of tract damage, the advantage here is to deal with a measure with greater biological value. To avoid selecting tracts to be analyzed on an arbitrary basis and to maintain statistical power, we included only tracts if they were damaged in at least 25% of patients at a proportion of at least 5%. Accordingly, the following 25 tracts were subjected to analysis (in parenthesis, the proportion of patients with at least 5% of the fibers interrupted): anterior commissure (33.6%), arcuate fasciculus (53.1%), the four strata of the cingulum bundle, including the fronto-parietal (33.6%), fronto-parahippocampal (31.1%), para-hippocampal (25%) and parahippocampal-parietal (27.4) pathways, three parts of the corpus callosum, including the anterior (49.21%), mid-anterior (52.3%) and posterior (41.4%) the frontal cortico-pontine tract (41.4%), the anterior (48.4%) and superior (44.5%) fronto-striatal tract, the extreme capsule (50.8%), the fornix (27.34%), the frontal aslant tract (48.4%), the inferior fronto-occipital fasciculus (50.8%), and the inferior longitudinal fasciculus (42.2%), the middle longitudinal fasciculus (28.9%), layers I (28.7%), II (54.7%) and II. (27.3%) of the superior longitudinal fasciculus, the anterior (44.5%) and superior (47.6%) parts of the thalamic radiations and the uncinate fasciculus (48.4%). Non-parametric Spearman correlations were performed between disconnection estimates and behavioral measures. A Bonferroni correction was applied to control for the number of tracts to be analyzed (critical *p*-value = 0.002). Note that more complex analyses to model the impact of cerebral disconnections on behavioral measurements of visuo-spatial attention were not performed in view of both the strong nonnormality and orthogonality of potential predictors.

**Case study**. In a separate analysis, we analyzed the case of a patient who has benefited from a two-step, sequential neurosurgery to remove a tumor invading the right cingulum. To access the anterior part of the tumor, the first surgery selectively targeted the pre-SMA and the caudal part of the anterior cingulate as well as, laterally, the postero-dorsal part of SFG. To access the posterior part of the tumor, the anterior part of both the SPL and precuneus was removed 6 months later. To better identify the cortical structures damaged by these two successive neurosurgeries, a pial mesh of the patient's brain was reconstructed with BrainVISA/anatomist (https://brainvisa.info, version 4.6). In addition, the cerebral disconnections caused by the surgical procedure were estimated with LQT (see above). The patient benefited from an assessment of visuo-spatial attention in the context of both surgeries.

**Statistics for behavioral analyses**. In this study, we used parametric statistics to assess between-group and within-subject differences (i.e., *t*-tests, repeated-measure ANOVAs, and mixed ANOVAs); all these tests were two-tailed and a *p*-value of less than 0.05 was considered as statistically significant. Post-hoc, pairwise multiple comparisons analyses were conducted with the Scheffé test which controls for the familywise error rate.

Parametric statistics were preferred despite the typical nonnormality of neuropsychological data because of the large sample size and to allow the assessment of interaction effects. Note that all analyses were also performed with the corresponding non-parametric statistics (except for the interaction effects related to mixed ANOVAs). The results were strictly the same. For the sake of

completeness, these analyses are provided in the supplementary information file (Supplementary Tables 8–11).

**Reporting summary**. Further information on research design is available in the Nature Research Reporting Summary linked to this article.

## Data availability

The unthresholded SVR-LSM *p*-maps, the raw beta maps, the raw lesion maps, and the overlap maps generated in this study as well as the raw behavioral dataset are available without restriction at https://doi.org/10.6084/m9.figshare.16822306.v3[79] All raw behavior data are also provided in the Source data files. The 1065-HCP tractography atlas is available at https://brain.labsolver.org/. Source data are provided with this paper.

## Code availability

SVR-LSM analyses were performed with the Matlab code by Zhang et al. (https://github.com/yongsheng-zhang/SVR-LSM). Optimization of SVR-LSM hyper-parameters was performed with the Matlab code by Wiesen et al. (https://data.mendeley.com/datasets/2hyhk44zrj/2). Measures of disconnection severity were performed with Lesion Quantification Toolkit[77] (https://wustl.box.com/v/LesionQuantificationToolkit). Source data are provided with this paper.

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

## Author contributions

G.H. and H.D. both contributed to the conception of the work, and to the acquisition and interpretation of data. G.H. performed all analyses and wrote the paper. H.D. provided a critical review of the manuscript. G.H. and H.D. approved the submitted version of the manuscript.

## Competing interests

The authors declare no competing interests
