## [Peer Review File · Nature Communications]

Contribution of the medial eye field network to the voluntary deployment of visuospatial attentionREVIEWER COMMENTS

Reviewer #1 (Remarks to the Author):

Thank you for the opportunity to comment on this elegant study by Herbet & Duffau. The authors report an interesting observation of partially dissociated changes in performance on two aspects of visuospatial attention/neglect in a large population of patients undergoing surgery for gliomas. Temporary performance change on a line bisection task was associated with surgeries limited more exclusively to dorsal parietal structures, while temporary declines in performance on the bell cancellation task arose from surgeries involving medial structures including SMA and middle cingulate cortex (as well as the same dorsal parietal structures). The study contributes novel and interesting data regarding the role of the medial cingulate and supplementary eye fields, data on which in general remains sparse. The methodology is reported in sufficient detail for replication and appears sound overall (see below for a question relating to registration). I have a few questions mainly around patient performance details:

How many individual patients showed evidence of visuospatial neglect prior to surgery? If none/very small number, could it not be that the absence of VLSM results with baseline performance might reflect insufficient variation in performance to detect an association, rather than reflecting functional reorganisation as proposed? The potential of reorganisation and the mass effect caused by glioma creates an interesting separate problem. How many patients had a tumour that involved the structures identified by VLSM as associated with aspects of neglect, but not have preoperatively any neglect symptoms?

In terms of the lesion distributions (Fig 1), please detail how many individual tumours / surgeries affected the main structures of interest (FEF, CEF, SEF).

For the behavioural analyses, how many individual patients showed symptoms of neglect in the clinically relevant range for each task? Where were tumours/surgeries located for clinically impaired patients in relation to FEF, SEF, CEF?

No pre- or post-operative tractography data were collected. Instead all disconnection analyses/interpretations are based on registering patients to a normal atlas indicating where tracts are expected to be. However, gliomas affect white matter tracts in different ways (sometimes growing around, sometimes displacing). Therefore, how certain can the authors be about the amount of 'tract disconnection' that occurred in this sample, especially since patients were operated awake to preserve tracts? This analysis is to my reading the main weak point of the study, and would benefit at least from explicit discussion.

In the fascinating single patient case who underwent repeat surgery, the first surgery involved primarily the medial structures associated with attentional aspects of neglect (assessed by the bell cancellation test). The VLSM analysis indicated that performance on this task draws also on the lateral parietal structures which were spared in the first surgery. During the second surgery, both attentional and perceptual aspects of neglect were identified, and additionally the symptoms were long-lasting, as opposed to the transient deficits seen in most patients undergoing single stage surgery. This is an intriguing result, since the patient initially recovered from 'attentional' neglect but then suffered it again (as well as 'perceptual' neglect) the second time. Did any of the surgeries in this series involve removing both the 'medial' and 'lateral' structures/systems in the same surgery, and did this cause longer-lasting symptoms at the individual level? It would be interesting to know the authors' thoughts about possible mechanisms. If the first recovery was due to functional reorganisation to contralateral structures, it seems unexpected for the same deficit to re-occur after second surgery in the same hemisphere. Would this perhaps indicate that residual white matter connections after the first surgery allowed for recovery, but remnant fibres were then damaged more extensively in the second surgery (see above comment about lack of certainty that fibres were "fully "disconnected" without tractography)? If so, could an alternative interpretation be that the cumulative damage to a shared network of tracts is relevant to both the nature and duration of symptoms, rather than

indicating a strong dissociation at the level of the white matter? Along these lines, the new correlation between line bisection and left_bell cancellation at delta 1 and delta 2 (supplementary table 3) could be interpreted to indicate that surgeries affected both functional systems together to some extent.

Small methodological clarifications:

Did the registration algorithm take account of the tumour masks and resection cavities to avoid registration bias (by excluding tumour/cavity voxels)?

Please add p-values to the significant findings in bold font in supplementary Table 3.

It would seem relevant to briefly highlight the known interplay between attentional orientation and perceptual aspects of attention as a likely/possible substrate for the shared reliance of bells cancellation test on both 'medial' and 'dorsal' structures (for example any of numerous works by Stokes / Nobre).

Thank you for presenting these interesting data.

Reviewer #2 (Remarks to the Author):

Herbet and Duffaut mapped the anatomy of spatial neglect in tumor patients before, immediately after and later after resection. Neglect was measured by line bisection and a cancellation task. Anatomic-behavioral correlations were mapped either on voxel-level by multivariate lesion symptom mapping or on tract-wise disconnection-level by correlation analysis. They found a role of the human medial eye field network with different relations to the 2 utilised tasks.

The study's major strength is the large sample of tumor patients with a longitudinal design. Stroke is indeed limited in mapping medial areas, while the sample in the current study covers brain areas that are not commonly damaged by stroke. Therefore, the study provides new insights into the anatomy of spatial neglect. Further, the longitudinal design allows circumventing some of the limitations of mapping tumor.

All in all, the study is well-justified and interesting. However, before I can suggest publication, I see the need for some revisions. There are some methodological peculiarities that should be clarified/corrected, and the discussion of results in the context of previous studies should be extended. Generally, the methods are described very minimalistically and I would appreciate more details to ensure that the study is replicable and its methodological rationale is understandable. On the other hand this might just be the common method writing style in high-impact journals, and the present section might follow the journal's guidelines. See below for a list of major and minor comments.

Major

1) Many discussions in the lesion mapping field circle around the question what etiologies should be used. I believe that this whole discussion is mostly just a highly biased rationalization of each researcher's own work, which depends on what resources are available to her/him at all. In the end, all etiologies come with certain disadvantages. This is also the case for tumors, which were the sole etiology in the SVR-LSM on time point A1. It is tricky to identify damaged tissue in tumors and due to a tumor's slow growth, brain anatomy can and will adapt, hence the mapping analysis will not exactly map healthy human brain anatomy anymore. This is only a minor issue with the study's sophisticated longitudinal design. As long as it is transparently mentioned as a limitation, I am fine with this. On the other hand, the analysis at A1 found no significant results for the bells test. This is explained by the "brain's efficient abilities to reorganize in response to glioma progression", which acknowledges the method's limitation. However, there are other explanations - Maybe the SVR was simply unable to model the target variable (see also my point below on SVR) or the patients simply did not show any pathological variance to be modeled. It would be quite important to show that patients suffered from deficits at A1, therefore a comparison to a control sample or established cutoffs would be helpful. It

must be shown that there were some patients with a deficit in the sample, else the whole A1 SVR-LSM might be considered meaningless. The same is the case for the $\Delta 2$ analysis. If there are no or hardly any patients at A1 and A3 with a clinical deficit, a mapping of the difference is quite meaningless.

2) Could you explain your choice to analyze right and left omissions in the bell test separately?

Usually, spatial neglect is defined as a lateralised attention deficit. Hence, I believe that it would be more informative to investigate the right-left difference or to use any other measure that assesses the lateralised bias of attention (e.g. <https://doi.org/10.1016/j.neuropsychologia.2010.04.018>). If you do not have a good reason for the design, I'd suggest adding an additional analysis with such measure as the main analysis of the cancellation test.

3) The image normalization procedure appears odd. You only report the use of "default normalization parameters" in SPM. However, as with all brain pathology visible in the scan, the pathology will interfere with the normalization process and potentially induce systematic errors (see e.g. PMC2938189; PMC2658465). Could you specify their procedure in detail? Note that cost function masking or enantiomorphic normalization will work not only on stroke lesions, but on tumors as well. If you did not use such strategy, I'd like you to provide some additional results to demonstrate good normalization. This could be, for example, a set of the first 15 patient images pre- and post-normalization.

4) In the introduction, you claim that the multivariate "SVR-LSM [has] a greater biological value since the method takes into consideration the inter-dependent nature of the voxels forming a given lesion map and is in principle better suited to control for the non-stochastically spatial distribution of brain damage" This claim has been refuted by several previous studies (e.g.

<https://doi.org/10.1016/j.neuropsychologia.2010.04.018><https://pubmed.ncbi.nlm.nih.gov/30549154/>,
<https://onlinelibrary.wiley.com/doi/full/10.1002/hbm.25278>)

5) You used default hyperparameters for the SVR-LSM. This is per se a suboptimal procedure – we never know how well hyperparameters transfer from the sample in Zhang et al (who also used different parameters across analyses!) to your sample and deficit. This is even a more severe issue with the current sample, as hyperparameters were originally optimized on stroke data by Zhang et al.. However, given the tricky situation with model fit and reproducibility as opposing criteria in SVR-LSM (Zhang et al., 2014), I could accept the use of the default parameters. In this case, I would like you to shortly mention this limitation and to report both values – out-of-sample model fit and reproducibility – for the chosen parameters. This is quite important to evaluate the quality of the SVR. Only if the SVR explains at least some behavioral variance, the mapping results can be considered reliable at all.

6) A main conclusion is that the common concept of a parietal correlate of line bisection and a frontal correlate of cancellation tasks are only part of the story, and the current study adds to it. However, I believe this summary of the current state to fall far too short. There are many studies that assessed neglect by cancellation and found parietal, temporal, and subcortical correlates. Of note, even Verdon et al. did not find exclusively frontal/exclusively parietal correlates, but always both to different degrees (see Figure 5). Given the issues with classic univariate VLSM that you mentioned, this might well be a methodological artifact. Please provide a more nuanced discussion of previous neglect studies (e.g. [10.1016/j.cortex.2011.11.008](https://doi.org/10.1016/j.cortex.2011.11.008); [10.1093/cercor/bhh076](https://doi.org/10.1093/cercor/bhh076);

<https://doi.org/10.1016/j.neuropsychologia.2008.08.020>; [10.1093/brain/awg200](https://doi.org/10.1093/brain/awg200) and many more)

7) I have some doubt about the single patient. If I understood correctly, you did the same disconnection analysis in the LQT Toolbox for this single patient. If I remember correctly, the toolbox uses probabilistic maps of WM tracts. This might not be problematic with large samples, but a strong conclusion from a single case is on thin ice. You should carefully discuss what kind of data the analysis is based on (i.e. is it really probabilistic?) and discuss this as a limitation given considerable variance in the anatomy of fiber tracts. Regarding this variance, see e.g. the atlas by Zhang et al ([10.1016/j.neuroimage.2010.05.049](https://doi.org/10.1016/j.neuroimage.2010.05.049)), where the frontoparietal part of the SLF is in no voxel more likely to be present than ~80%, hence we can hardly ever guarantee affection of the SLF-FP in any single patient.

Minor

1) I don't really understand the purpose of Background analyses, the first section of the results. You state that you wanted "to gauge the weight of [demographic] variables on the behavioral measurements of visuo-spatial attention." What is the 'weight' of a demographic variable? And why do

you include tumor size/resection volume in the analysis? In the SVR-LSM, you perform a regression of brain damage location (not size!) on symptoms. Hence, it doesn't look to me like you mean it serious with the factor tumor size/resection volume. Maybe age just explains variance in your analysis because resection volume isn't such a good predictor, and voxel-wise (/tract-wise) anatomical data would explain the data, leaving nothing to age. I would prefer a simple descriptive analysis.

2) In the section "patient sample", a small side remark states that no patients suffered from visual extinction. This is quite interesting, given that some resections included parietal areas. How and when was extinction tested? It would be great if you could provide some more information on this point.

3) Regarding correction for multiple comparisons, you state that "A FDR correction ($q = 0.05$) estimated by a bootstrap permutation procedure (5000 permutations) was used to threshold the resulting SVR-LSM β -maps". This is quite imprecise. There is first a permutation to estimate the significance of the voxel's feature weight, and then FDR is applied. Anyway: Were all methods applied inside the SVRLSM scripts from Zhang et al? Please state. Could you cite the applied FDR method?

4) Group analyses: the ANOVA showed "most importantly a powerful interaction". The term 'powerful' is imprecise here. It's highly significant, that's all. As the term 'statistical power' exists, this might be confusing.

5) In the discussion, you interpret p-values from correlation analysis by their magnitude: some "were clearly more significant". Aside from the problems with a comparison of significance levels in inferential statistics, why don't you just look at the correlation values, which tell you something about the actual strength of a relation between two variables?

Reviewer #3 (Remarks to the Author):

This is a potentially interesting paper: nonetheless some clarifications and rewriting seems mandatory at the moment. In the following I have detailed a number of points that the authors might wish addressing in revision

Intro page 3: please note that there is growing evidence that a strict dorsal-ventral distinction between endogenous/orienting and exogenous/re-orienting control, respectively, might be incorrect because in response to invalid targets: 1) the ventral TPJ area is in charge of the late processing of the match/mismatch between the cued and actual target location for updating the contextual probabilistic association between cue direction and target location (see Doricchi et al., 2010; Geng and Vossel, 2013; Macaluso and Doricchi, 2013); b) the early operation of re-computing the direction of attention to the actual target location, i.e. spatial reorienting, is played by the superior rather than inferior parietal lobe (Vandenberghe et al., 2012; Ptak and Schnider, 2011): note that the findings reported by the authors support this view. The authors might wish to smooth down the old-fashioned dorsal/ventral dichotomy.

Page 2 Intro:

"However, almost nothing is known about the contribution of these medial areas ..." may be it would be better that " little is known " (see for example ref. 56, 57). Same when later on the authors state "...neuropsychological studies have consequently failed in delineating the exact role of the dorsomedial structures "

Page 5 Intro:

"While the association between SLF II and III and spatial neglect has been experimentally evidenced^{25,26,49}, the causal role of SLF I remains elusive.... ", at least partially elusive: ref 56 reported the effect of SLF I disconnection.

Page 11: "In summary, the above analyses confirmed that the bell test was affected to the same extent by resections targeting either the parietal or the fronto-medial areas. By contrast, line bisection performances were uniquely impaired following parietal resections. This means that resections of the medial frontal lobe (especially SMA and the adjacent middle/anterior cingulate cortex) specifically

impacted left_bell." I'm not fully convinced by the logic of this sentence: these results suggest that parietal resections specifically impacted line bisection whereas both parietal and medial frontal lobe resections impacted left bell. The authors might wish to rewrite or drop this paragraph.

Page 17: "Our findings clearly demonstrate that damage to these two eye-related areas is able to produce visuo-motor exploratory neglect, though the underlying mechanisms cannot be fully elucidated here; it might be either the consequence of a deficit in initiating saccades towards the contralesional space, in line with the established behavioral affiliations of the medial frontal structures⁵⁷, or the result of specific difficulties in deliberately orienting attention during visual search (i.e. the attentional function attributed to the FEF). The latter interpretation might be more likely, as ablation or inactivation studies in monkeys have generally shown not or only little impact of SEF/CEF damage on saccade initiation^{54,58}. Another possibility is that visuo-motor exploratory neglect arises from a functional diaschisis effect⁵⁹ transiently impairing the FEF by deprivation of functional inputs – the FEF, SEF and CEF being densely interconnected^{19,20}." On the one hand I'm very positively impressed by the scholarly compilation of such an exhaustive set of hypotheses though, on the other hand, some of these hypotheses weaken significantly the strength of the main conclusion of this paper, i.e. that it provides first evidence for a role of the frontal-medial network in the voluntary deployment of visuo- spatial attention. Put in other words, although the study is methodologically sound and was scholarly run on an extended samples of carefully clinically selected patients with low-grade brain gliomas, the same study looks like providing new interesting insights in the neglect syndrome though not final evidence on the role played by the frontal-medial network. I think the authors might wish to "positively" smooth the tone of their conclusions and treat the empirical evidence they have gathered as importantly pointing at the possible role of medial frontal-parietal structured in the guide of attention.

References.

Doricchi, F., Macci, E., Silvetti, M., & Macaluso, E. (2010). Neural correlates of the spatial and expectancy components of endogenous and stimulus-driven orienting of attention in the Posner task. *Cerebral Cortex*, 20(7), 1574e1585.

Geng, J. J., & Vossel, S. (2013). Re-evaluating the role of TPJ in attentional control: contextual updating? *Neuroscience & Biobehavioral Reviews*, 37(10), 2608e2620.

Macaluso, E., & Doricchi, F. (2013). Attention and predictions: control of spatial attention beyond the endogenous- exogenous dichotomy. *Frontiers in Human Neuroscience*, 7.

Ptak R, Schnider A (2011) The attention network of the human brain: relating structural damage associated with spatial neglect to functional imaging correlates of spatial attention. *Neuropsychologia* 49:3063–3070.

Vandenberghe R, Molenberghs P, Gillebert CR (2012) Spatial attention deficits in humans: the critical role of superior compared to inferior parietal lesions. *Neuropsychologia* 50:1092–1103

Response to reviewers' comments

Reviewer #1

General comment: Thank you for the opportunity to comment on this elegant study by Herbet & Duffau. The authors report an interesting observation of partially dissociated changes in performance on two aspects of visuospatial attention/neglect in a large population of patients undergoing surgery for gliomas. Temporary performance change on a line bisection task was associated with surgeries limited more exclusively to dorsal parietal structures, while temporary declines in performance on the bell cancellation task arose from surgeries involving medial structures including SMA and middle cingulate cortex (as well as the same dorsal parietal structures). The study contributes novel and interesting data regarding the role of the medial cingulate and supplementary eye fields, data on which in general remains sparse. The methodology is reported in sufficient detail for replication and appears sound overall (see below for a question relating to registration). I have a few questions mainly around patient performance details:

The authors: Thank you very much for both your positive appreciation of the work and your thoughtful suggestions. We hope that the add-ons described in the following will be satisfying.

Major comment #1:

- (1) How many individual patients showed evidence of visuospatial neglect prior to surgery? If none/very small number, could it not be that the absence of VLSM results with baseline performance might reflect insufficient variation in performance to detect an association, rather than reflecting functional reorganisation as proposed?

The authors: To adequately address the issue pointed by the reviewer, we included a group of 44 healthy participants matched in terms of age, educational level and sex. The performances of this group were contrasted to those of the patient group at each time point (A1, A2 & A3) and for each behavioral measure of interest (i.e. *line bisection* estimates, *total_bell* and *diff_bell*). The data distributions of this control group was also used to compute individual Z-scores and thus to determine the proportion of deficits at each assessment. Note that *diff_bell* is a new dependent variable used on the request of Reviewer 2 (i.e. the asymmetry score, *left_bell* minus *right_bell*, which is in principle better suited for assessing the lateralized aspect of visuo-spatial attention; note that *left_bell* and *right_bell* are no longer used in the main corpus of the work). Please, see the following changes:

- In the section 'Patient sample' (Materials and Methods):

“ A control group of 44 neurologically healthy participants matched in terms of age, education level and sex was recruited to assess difference with the patient group (see Supplementary Table 4).”

- In the new section 'Behavioral analyses: patients versus healthy control participants' (Materials and Methods):

“**Behavioral analyses: patients versus healthy control participants.** To determine whether patients already showed a right bias in visuo-spatial attention before surgeries were performed, preoperative baseline performances (A1) were statistically compared to those gained from a healthy control group (44 participants) matched in terms of age ($t_{(170)} = -1.13, p = 0.26$), education ($t_{(170)} = 1.06, p = 0.29$) and sex ($\chi^2 = 0.14, p = 0.71$) (see Supplementary Table 3 and 4 for details about the

behavioral data of both the patient group and the control group, respectively). A group effect was observed for *line bisection* estimates ($t_{(170)} = -3.45, p = 0.0007$), *total_bell* ($t_{(170)} = -2.32, p = 0.02$) but not for *diff_bell* ($t_{(170)} = 0.48, p = 0.63$). The same analyses were performed considering A2 and A3. In brief, all behavioral measurements were strongly different between groups at A2, and the same pattern than that observed at baseline was identified at A3. All statistical analyses are described in Supplementary Table 5 and Supplementary Figure 1. The frequency of individual deficits estimated from the normative distributions are shown in Supplementary Figure 2.”

- A supplementary table showing the sociodemographic characteristics of the control group:

Supplementary Table 4: demographic and behavioral data for the 44 healthy participants

	mean	SD	range
age	38.34	11.26	[19; 68]
education (full-years)	15.27	2.67	[19; 20]
sex	20 Females; 24 Males		
handedness	all right-handed		
line bisection	-2.41	2.77	[-7.3; 4.3]
total_bell	1.22	1.51	[0; 5]
diff_bell	0.16	1.27	[-3; 4]

- A supplementary table detailing the between-group statistical analyses:

Supplementary Table 5: Comparisons between controls and patients at each assessment

Time point	Measurements	$t_{(170)}$	p	diff in mean	- IC(95)	+IC(95)
A1	bisection line	-3.45	0.0007	-1.43	-2.25	-0.61
	total_bell	-2.32	0.021	-0.97	-1.79	-0.15
	diff_bell	0.40	0.63	0.12	-0.37	0.61
A2	bisection line	-3.74	0.00025	-3.35	-5.12	-1.58
	total_bell	-5.43	1.90^{e-07}	-4.37	-5.95	-2.78
	diff_bell	-3.43	0.0008	-2.19	-3.46	-0.93
A3	bisection line	-2.81	0.006	-1.61	-2.73	-0.48
	total_bell	-2.15	0.033	-0.81	-1.56	-0.07
	diff_bell	-0.23	0.82	-0.06	-0.56	0.44

Two-tailed *t*-tests were used to determine statistical difference between the control and the patient group for each behavioral measurement and for each assessment. Note that non-parametric statistics (Mann-Whitney) lead to the same results (see also **Supplementary Figure 1**).

- A new supplementary figure providing density plots of healthy controls *versus* patients:

Supplementary Figure 1: Density plots showing the data distribution of healthy participants (in green) *versus* patients at each time point (A1, A2 & A3) for *line bisection* estimates (A), *total_bell* (B) and *diff_bell* (C). Vertical lines represents the mean of each distribution. See Supplementary Table 5 for details about the statistical analyses. A1, preoperative assessment; A2, 5-day postoperative assessment; A3, three-month postoperative assessment. R software (<https://www.R-project.org/>; packages = *ggplot2* & *ggpubr*) was used to create this figure.

- A new supplementary figure showing the proportion of deficits at each time point and for each behavioral measurement:

Supplementary Figure 2: Proportion of individual deficits at each assessment for the three behavioral measurements of interest. Z-scores were computed based on the normative distributions of healthy participants. A score was considered as impaired when the corresponding z-score was equal or superior to 1.65 ($p = 0.05$ one-tailed). As shown in this figure, the rate of deficits at A1 was very close to what it would be expected in the normal population (i.e. 5%), especially for *line_bisection_estimates* and *diff_bell* – the two direct measures of neglect in this study. R software (<https://www.R-project.org/>; packages = *ggplot2* & *ggpubr*) was used to create this figure.

To summarize, these additional results showed that there were rather small or no differences (depending on the behavior measure considered) between the control group and the patient group at preoperative baseline A1. This suggests that there was not enough pathological variance to be modeled by SVR-LSM at A1, in consistence with the hypothesis of preoperative functional compensation we proposed (and which is widely documented in the context of lower grade glioma¹⁻³). If such functional compensation did not occur preoperatively, it would have been expected greater between-group differences and a larger proportion of individual deficits. We call the reviewer attention to the fact that, in the revised version of the manuscript, hyper-parameters of SVR-LSM models have been now optimized *via* grid search (please see our complete response to *the reviewer 2' major comment 5*). We failed to find a combination of hyper-parameters (i.e. C and γ) associated with a satisfactory goodness-of-fit and/or reproducibility for the three behavioral measures at A1. This is another argument to say that there was not enough pathological variance to be modeled by SVR-LSM at this time point. As a result, no SVR-LSM models were generated at preoperative baseline.

1 Duffau, H. Lessons from brain mapping in surgery for low-grade glioma: insights into associations between tumour and brain plasticity. *The Lancet Neurology* **4**, 476–486 (2005)

2 Desmurget, M., Bonnetblanc, F. & Duffau, H. Contrasting acute and slow-growing lesions: a new door to brain plasticity. *Brain* **130**, 898–914 (2007).

3 Herbet, G., Maheu, M., Costi, E., Lafargue, G. & Duffau, H. Mapping neuroplastic potential in brain-damaged patients. *Brain* **139**, 829–844 (2016).

-
- (2) The potential of reorganisation and the mass effect caused by glioma creates an interesting separate problem. How many patients had a tumour that involved the structures identified by VLSM as associated with aspects of neglect, but not have preoperatively any neglect symptoms?

The authors: Thank you for this suggestion that makes sense. We created a new table to show these data. We calculated the proportion of patients with a deficit at A1 for both the parietal group and the MFC group (Parietal and MFC areas being the two main loci identified by SVR-LSM for delta_1), and the proportion of new deficits at A2 (the proportion of patients that did not show a deficit prior to the surgery). Please see the following Supplementary Table.

Supplementary Table 7: Proportion of parietal and MFC patients with a new deficit after surgery

	Parietal Group				MFC Group			
	deficit A1		new deficit A2		deficit A1		new deficit A2	
	n	%	n	%	n	%	n	%
Bisection line	1	5.5	11	61.1	1	4.7	3	14.3
total_bell	3	16.7	13	72.2	3	14.3	16	76.2
diff_bell	1	5.5	8	44.4	1	4.7	14	66.7

In this table, it is shown the proportion of patients with a pathological deviation at A1 ($z\text{-score} \geq 1.65$) and the proportion of patients with a new deficit at A2 (i.e. patients for whom the performance was unimpaired prior to surgery). The proportion of new deficits is relatively comparable between both groups for *total_bell* (2-tailed $\chi^2 = 0.08$, $p = 0.78$) and *diff_bell* (2-tailed $\chi^2 = 1.95$, $p = 0.78$), but unequal for *line bisection* estimates (2-tailed $\chi^2 = 9.23$, $p = 0.0024$). This is in agreement with the results from the between-group analyses described in the main text.

Major comment #2: In terms of the lesion distributions (Fig 1), please detail how many individual tumours / surgeries affected the main structures of interest (FEF, CEF, SEF).

The authors: We thank the reviewer for this suggestion. Unfortunately, it is difficult to provide an accurate value (at least for the CEF/SEF), just an estimate. Both areas are not anatomical ones and, ideally, localizer fMRI should be used to detect the CEF and SEF on a subject-by-subject basis, which is of course impracticable in patients for whom the areas in question are overwhelmed by the tumour. According the current literature, the SEF is generally defined as located at the junction between the pre-SMA and the SMA, and the CEF at the junction between the middle cingulate and the anterior cingulate - these definitions can slightly vary across studies and depend on the behavioral tasks used to fire their neural activities (e.g. 1,2). Regarding the FEF, it is generally defined as located at the junction between the precentral and the superior frontal sulcus by fMRI studies or as located in the most posterior and dorsal portion of the middle frontal gyrus by electrostimulation studies

(e.g. 3). Note that the FEF was not an area of interest in this study because poorly affected by neurosurgeries (this is indicated in the section 'lesions distributions').

In the revised version of Figure 1, the relative location of both the CEF and SEF is indicated. In the figure legend, an estimation of the number of tumors or resection cavities involving both areas are given. On the Figure, the reviewer can see that the FEF is poorly infiltrated and poorly resected (a new arrow indicates the typical location of the FEF). Please see the changes:

Figure 1: Density overlap map of (A) preoperative tumors and (B) surgical cavities for the 128 patients included in this study. These maps are thresholded in such a way that only voxels affected in at least three patients (the threshold for SVR-LSM analyses; see Materials and Methods) are shown. Bars indicate lesion density. The maximum overlap was in the insula for both the tumor map ($n = 52$) and the resection cavity map ($n = 40$). **The relative location of the supplementary eye field (SEF) and the cingulate eye field (CEF) is indicated. For indicative purposes, both areas were more or less infiltrated in 16 patients before surgery and further more or less resected in 18 patients.** AG, angular

gyrus; FEF, frontal eye field; IFG, inferior frontal gyrus; PreC, precuneus; Pre-SMA, pre-supplementary motor area; SFG, superior frontal gyrus; TP, temporal pole.

1 Pierrot-Deseilligny, C., Müri, R. M., Ploner, C. J., Gaymard, B. & Rivaud-Péchéux, S. Cortical control of ocular saccades in humans: a model for motricity. *Progress in brain research* **142**, 3–17 (2003).

2 Müri, R. M. MRI and fMRI analysis of oculomotor function. *Progress in brain research* **151**, 503–526 (2006).

3 Vernet, M., Quentin, R., Chanes, L., Mitsumasu, A. & Valero-Cabré, A. Frontal eye field, where art thou? Anatomy, function, and non-invasive manipulation of frontal regions involved in eye movements and associated cognitive operations. *Frontiers in integrative neuroscience* **8**, 66 (2014).

Comment #3: For the behavioural analyses, how many individual patients showed symptoms of neglect in the clinically relevant range for each task? Where were tumours/surgeries located for clinically impaired patients in relation to FEF, SEF, CEF?

The authors: The proportion of patients with a deficit for each behavioral measure and for each time point is given in the new Supplementary Figure 2. The proportion of deficits at A1 and the proportion of new deficits at A2 for the MFC group (involving the CEF and the SEF) are given in Supplementary Table 7 (see our complete response to comment #1). Note that we cannot segregate patients in which the CEF or the SEF was specifically removed since in the vast majority of patients with a fronto-mesial resection, the pre-SMA/SMA and anterior/middle cingulate are surgically removed. This is the reason why we used the inclusive term “medial network” or “medial eye field network” in the discussion.

Comment #4: No pre- or post-operative tractography data were collected. Instead all disconnection analyses/interpretations are based on registering patients to a normal atlas indicating where tracts are expected to be. However, gliomas affect white matter tracts in different ways (sometimes growing around, sometimes displacing). Therefore, how certain can the authors be about the amount of ‘tract disconnection’ that occurred in this sample, especially since patients were operated awake to preserve tracts? This analysis is to my reading the main weak point of the study, and would benefit at least from explicit discussion.

The authors: The comment of the reviewer is right, but especially applies to higher grade glioma, much less to diffuse low-grade glioma (the tumor mainly propagates along the white matter connectivity with no or only slight mass effect). This is the reason why it is of importance to include homogenous patients, as this study – all the more since the pathophysiological mechanisms leading to deficits are not same between both classes of tumors. Moreover, disconnection analyses were performed on the basis of *resection cavity maps*, not on the basis of preoperative tumor maps – considerably limiting the potential bias related to possible tract displacement (release of mass effect if any). If such displacements nevertheless occurred in some patients, the sample size (which is relatively large) allows to smooth the effect of this potential bias on the final results.

This is also true that patients were operated on under awake condition allowing to preserve some critical white matter connections; however, all tracts are of course not preserved (only tracts associated with a function/process monitored with a particular task during awake surgery). This is typically the case of SLF I and cingulum which are most of the time surgically removed (please see Herbet et al., 2016, *Brain*, Figure 4 & Figure 6; Please see also Figure 1A of the current work). Moreover all fibers of a same tract are not spared.

Importantly, and as clearly stated in the manuscript, spatial cognition was uniquely assessed intra-operatively (with the line bisection task) when the tumor was located in the parietal cortex, not in the frontal one. In the manuscript, it is explicitly acknowledged that the sample is not unbiased and that some tracts expected to be more significantly involved in postoperative neglect (in particular the SLF_II) are probably not because spared at least partially during the surgical procedure:

“Third, the patient sample on which we relied is not unbiased as, for some of them, an intraoperative mapping of visuospatial attention was performed in an attempt to safely remove tumors lodged in the posterior parietal cortex. This may explain for example why we only found a limited and transitory effect for SLF_II which is typically mapped and spared to avoid lasting postoperative neglect^{4,5}. However, this does not apply to regions outside the parietal cortex for which visuospatial attention was not intraoperatively monitored.”

Last, as indicated in the materials and methods section, tracts were subjected to analyses if they were damaged in a proportion of 5% in at least 25% of patients.

To mitigate the concern pointed by the reviewer, we added a Supplementary Note in the revised version of the work.

- Supplementary Note 3:

Supplementary Note 3. Disconnection analyses were performed in this study to ascertain the extent to which surgical resections could account for the occurrence of neglect signs in the short and longer term. A note of caution should be clearly mentioned here. The patient sample on which we relied was uniquely composed of patients harboring a lower-grade glioma. Although compared to higher grade glioma mass effect is much less frequent in this tumor grade, it may nevertheless occur in certain patients and consequently bias the expected spatial positioning of white matter tracts (and thus the measures of disconnection severity). However, we are relatively protected from this potential shortcoming as disconnection analyses were uniquely performed on the basis of resection cavity maps (if any, the mass effect is released at this stage). Moreover, the sample size – which is relatively large - allows to smooth, at least to some extent, the effect of this potential bias on the final results.

Comment #5: In the fascinating single patient case who underwent repeat surgery, the first surgery involved primarily the medial structures associated with attentional aspects of neglect (assessed by the bell cancellation test). The VLSM analysis indicated that performance on this task draws also on the lateral parietal structures which were spared in the first surgery. During the second surgery, both attentional and perceptual aspects of neglect were identified, and additionally the symptoms were long-lasting, as opposed to the transient deficits seen in most patients undergoing single stage surgery. This is an intriguing result, since the patient initially recovered from ‘attentional’ neglect but then suffered it again (as well as ‘perceptual’ neglect) the second time. Did any of the surgeries in this series involve removing both the ‘medial’ and ‘lateral’ structures/systems in the same surgery, and did this cause longer-lasting symptoms at the individual level? It would be interesting to know the authors’ thoughts about possible mechanisms. If the first recovery was due to functional reorganisation to contralateral structures, it seems unexpected for the same deficit to re-occur after second surgery in the same hemisphere. Would this perhaps indicate that residual white matter connections after the first surgery allowed for recovery, but remnant fibres were then damaged more extensively in the second surgery (see above comment about lack of certainty that fibres were ‘fully “disconnected”’ without tractography)?

If so, could an alternative interpretation be that the cumulative damage to a shared network of tracts is relevant to both the nature and duration of symptoms, rather than indicating a strong dissociation at the level of the white matter? Along these lines, the new correlation between line bisection and left_bell cancellation at delta 1 and delta 2 (supplementary table 3) could be interpreted to indicate that surgeries affected both functional systems together to some extent.

The authors: We thank the reviewer for this valuable comment and interpretation. Let's us clarify some points.

First, we would like to reiterate that the performances related to both behavioral tasks cannot be fully dissociated in the context of perceptual neglect, but only in the context of visuo-motor exploratory neglect (this is the hypothesis developed in this study, please see the fourth paragraph of the introduction section: "*However, a strict double dissociation is unlikely because target cancellation performances can be affected by perceptive neglect.*") (P5). This implies that patients with a perceptive neglect will necessarily show impaired performances on both tasks (i.e. the visual field is indeed neglected on the left side), while patients with a visuo-motor exploratory neglect will be only impaired on a visual search task such as the bell test (patients are no longer able to deliberately explore the contralateral space). In our opinion, this dissociation in performance is well illustrated in Figure 4: performances are clearly dissociated between the parietal and the MFC group for *line bisection* estimates, not for the bell test (*diff_bell* or *total_bell*). This pattern of results was replicated at the individual level in the patient who underwent a two-stage surgery. The first surgery led to a visuo-motor exploratory neglect (only the bell test was impaired), while the second one led to perceptive neglect (both line bisection and bell test were impaired). It is thus unlikely (we agree) that the same deficit re-occurred (this is not interpretation developed in this study).

The point related to the lack of recovery is a very interesting one, but the current data do not allow to provide a clear-cut interpretation of this long-lasting deficit which is likely to be multidetermined. In particular, we call the reviewer attention on the fact that no mapping of visuospatial cognition with the line bisection task was performed in this patient (as indicated in the section 'Materials and Methods') and that we used a trans-cortical approach through the SPL/precuneus to access the posterior part of the tumor. In this context, the perceptive neglect experienced by the patient may arise from both topological (i.e. cortical damage) and multiple disconnection mechanisms (following the second surgery, patients suffered from other disconnections than SLF_I including for example CC_midpost and EMC). Having said that, the hypothesis whereby neglect severity may be related to a "cumulative damage of shared network of tracts" remains plausible, but difficult to substantiate here given the results of the disconnection analyses (Please see our response to reviewer 2's comment #7 and the subsequent add-ons in which we toned down the statement related to the full "disconnection" of SLF_1 given the methods used to estimate disconnection severity).

Last, the shared variance between *diff_bell* (i.e. the asymmetry score i.e. the difference left-right bell) is rather low ($r = 0.23$ for A1, $r = 0.25$ for delta_1, $r = 0.14$ for delta_2) and mainly reflects the performances of parietal patients who show a deficit on both tasks). This result is expected since patients with a perceptive neglect are unaware of the left visual field.

We added a supplementary note to discuss the potential mechanisms leading to the long-lasting neglect, including that developed by the reviewer:

Supplementary Note 2. It is interesting to note that neglect symptoms were long-lasting in the single patient following the second surgery as opposed to transient (or significantly recovered) in most patients having undergone a single stage surgery. While the current data

does not allow to provide a clear-cut interpretation for this lack of recovery which is likely to be multidetermined, several lines of explanations can be offered. First, no mapping of visuo-spatial attention was performed with the line bisection task and we used a trans-cortical surgical approach through the anterior precuneus/SPL to access the posterior part of the tumor. Accordingly, the identified perceptive neglect might arise from a topological mechanism involving this cortical region. Second, beyond the SLF_I/ cingulum disconnection which was likely to already occur following the first surgery, damage to other tracts was observed, mainly including EMC and the middle-to-posterior part of the corpus callosum, and less severely the middle longitudinal fasciculus, parieto-pontine tract, medial lemniscus, posterior thalamic radiations and posterior cortico-striatal tract (see main text). As a result, a complex pattern of disconnection might also account for the lasting neglect. Last, the cumulative damage to the medial network (both cortically and subcortically) following both surgeries might severely diminish the possibility to initiate efficient strategies of functional reorganization, resulting in permanent neglect signs.

Small methodological clarifications

Minor comment #1: Did the registration algorithm take account of the tumour masks and resection cavities to avoid registration bias (by excluding tumour/cavity voxels)?

The authors: Cost function masking was initially not used because SPM12 performs very well without masking lesions during registration since its last major update. However, because the second reviewer also points this potential concern, we decided to register again all MRIs using enantiomorphic normalization (Nachev et al., 2008). This implies that all tumor/resection cavity maps were drawn again and consequently that all analyses were performed again with the new lesion masks. The results remain the same. Please see our complete response to reviewer 2' major comment 3.

Nachev, P., Coulthard, E., Jäger, H. R., Kennard, C. & Husain, M. Enantiomorphic normalization of focally lesioned brains. *Neuroimage* **39**, 1215–1226 (2008).

Minor comment #2: Please add p-values to the significant findings in bold font in supplementary Table 3.

The authors: Done.

Minor comment #3: It would seem relevant to briefly highlight the known interplay between attentional orientation and perceptual aspects of attention as a likely/possible substrate for the shared reliance of bells cancellation test on both 'medial' and 'dorsal' structures (for example any of numerous works by Stokes / Nobre).

The authors: Thank you very much for this suggestion. If the reviewer agrees, we prefer not discussing further on this possible interplay because our data do not really allow to support

this interesting hypothesis without being overtly speculative. In our opinion and as developed in the manuscript, it is not surprising that patient suffering from a perceptive neglect after parietal resection show a disturbance of both the line bisection task and the visual search task. Although visual search was also disturbed in the MFC group, the impaired mechanism does not appear to be the same. In other words, it seems difficult to us to consider the superior parietal lobule as a common area both forms of neglect and thus the locus for a dynamical interplay between the voluntary orientation of attention and the perceptive aspect of attention.

Reviewer #2

Reviewer's General comment: Herbet and Duffau mapped the anatomy of spatial neglect in tumor patients before, immediately after and later after resection. Neglect was measured by line bisection and a cancellation task. Anatomic-behavioral correlations were mapped either on voxel-level by multivariate lesion symptom mapping or on tract-wise disconnection-level by correlation analysis. They found a role of the human medial eye field network with different relations to the 2 utilised tasks.

The study's major strength is the large sample of tumor patients with a longitudinal design. Stroke is indeed limited in mapping medial areas, while the sample in the current study covers brain areas that are not commonly damaged by stroke. Therefore, the study provides new insights into the anatomy of spatial neglect. Further, the longitudinal design allows circumventing some of the limitations of mapping tumor.

All in all, the study is well-justified and interesting. However, before I can suggest publication, I see the need for some revisions. There are some methodological peculiarities that should be clarified/corrected, and the discussion of results in the context of previous studies should be extended. Generally, the methods are described very minimalistically and I would appreciate more details to ensure that the study is replicable and its methodological rationale is understandable. On the other hand this might just be the common method writing style in high-impact journals, and the present section might follow the journal's guidelines. See below for a list of major and minor comments.

The authors: We thank very much the reviewer for his/her thoughtful methodological comments and his/positive appreciation of the manuscript. As the reviewer will see, all suggestions have been taken in full consideration.

Major comment #1: Many discussions in the lesion mapping field circle around the question what etiologies should be used. I believe that this whole discussion is mostly just a highly biased rationalization of each researcher's own work, which depends on what resources are available to her/him at all. In the end, all etiologies come with certain disadvantages. This is also the case for tumors, which were the sole etiology in the SVR-LSM on time point A1. It is tricky to identify damaged tissue in tumors and due to a tumor's slow growth, brain anatomy can and will adapt, hence the mapping analysis will not exactly map healthy human brain anatomy anymore. This is only a minor issue with the study's sophisticated longitudinal design. As long as it is transparently mentioned as a limitation, I am fine with this. On the other hand, the analysis at A1 found no significant results for the bells test. This is explained by the "brain's efficient abilities to reorganize in response to glioma progression", which acknowledges the method's limitation. However, there are other explanations - Maybe the SVR was simply unable to model the target variable (see also my point below on SVR) or the patients simply did not show any pathological variance to be modeled. It would

be quite important to show that patients suffered from deficits at A1, therefore a comparison to a control sample or established cutoffs would be helpful. It must be shown that there were some patients with a deficit in the sample, else the whole A1 SVR-LSM might be considered meaningless. The same is the case for the $\Delta 2$ analysis. If there are no or hardly any patients at A1 and A3 with a clinical deficit, a mapping of the difference is quite meaningless.

The authors: As suggested, a control group (n = 44) was included to assess difference with the patient group but also to estimate the proportion of deficits at each time point and for the three behavioral measures of interest. Please see our comprehensive response to the reviewer 1' major comment 1 (a new section has been added in the main text, as well as two supplementary figures and two supplementary tables).

To summarize, our additional analyses and results indicated that there were rather small or no differences (depending on the behavior measure considered) between the control group and the patient group at A1 (*Supplementary Figure 1, Supplementary Table 5, and new section 'Behavioral analyses: patients versus healthy control participants' in the main text*). Likewise, the proportion of individual deficits was very low at A1, at least for *line bisection* estimates and *diff_bell* (difference left minus right) (*Supplementary Figure 2*). Furthermore, in the revised version of the manuscript, hyper-parameters of SVR-LSM models have been optimized *via* grid search (please see our response to comment #5) and we failed to identify a combination of parameters (i.e. C and γ) associated with a satisfactory goodness-of-fit and/or reproducibility for the three behavioral measures at A1 (thus SVR-LSM analyses were not performed). These results indeed suggest that there was not enough pathological variance to be modelled and are thus consistent with the view that a great deal of functional compensation occurred preoperatively in this patient group since greater differences or a larger proportion of deficits would have been expected if such compensation did not occur.

In the same way, in agreement with fact that patients regained in average their preoperative baseline performances (no significant differences between A1 and A3; $\Delta 2$ tends toward zero for the three measures), suitable parameters for $\Delta 2$ SVR-LSM models were not found (see comment #5).

Regarding the comment related to the brain's propensity to reorganize in response to glioma infiltration, this limitation is clearly mentioned and taken into consideration in the interpretations developed in the manuscript – as the reviewer acknowledges.

Major comment #2: Could you explain your choice to analyze right and left omissions in the bell test separately? Usually, spatial neglect is defined as a lateralised attention deficit. Hence, I believe that it would be more informative to investigate the right-left difference or to use any other measure that assesses the lateralised bias of attention (e.g. <https://doi.org/10.1016/j.neuropsychologia.2010.04.018>). If you do not have a good reason for the design, I'd suggest adding an additional analysis with such measure as the main analysis of the cancellation test.

The authors: We initially analyzed right and left omissions separately because neglect patients can suffer concomitantly from non-lateralized attention disturbances. As a consequence, using the asymmetry score (i.e. left/right difference) can lead to an underestimation of neglect severity in some cases. Having said that, we followed the reviewer' suggestion. As the reviewer will see, the main result of the SVR-LSM analysis remains the same, with however a higher degree of anatomical specificity for the new dependent variable.

In the main text, the analyses related to *left_bell* or *right_bell* have been suppressed and replaced by those related to *diff_bell*. Figures 3, 4 and 5 have been updated accordingly. Please see the following changes:

- In the section ‘Behavioral analyses: longitudinal performances’

“ Last, the asymmetry score (i.e. *diff_bell*) also evolved across the three assessments ($F_{(2, 254)} = 36.71$, $p = 1.44 \times 10^{-8}$, $\eta^2_p = 0.22$). Compared to the preoperative baseline, it was greater immediately after surgery (A1 vs A2; mean 0.04 ± 1.48 vs 2.35 ± 4.17 ; $p = 1.84 \times 10^{-10}$) but comparable three months later (A1 vs A3; mean 0.04 ± 1.48 vs 0.22 ± 1.52 ; $p = 0.86$) (Figure 2E). Accordingly, the difference between $\Delta 1$ (mean 2.31 ± 4.36) and $\Delta 2$ (mean -0.17 ± 2.07) was significant ($t_{(127)} = 5.60$, $p = 1.25 \times 10^{-7}$) (Figure 2F).

In summary, neurosurgeries impaired task performance, but only in the immediate postoperative period. For the bell test, items situated on the left side were considerably more affected than those placed on the right side (i.e. **significant increase of *diff_bell***) – a typical sign of spatial neglect.”

- In the section ‘SVR-LSM’ results:

“ [...] The SVR-LSM model for *total_bell* revealed a dissociated pattern of associations (Figure 3B): while areas of the parietal cortex were still identified, including the inferior and superior parietal lobules, the most significant cluster of suprathresholded voxels was detected on the medial face of the frontal lobe, including the supplementary motor area (SMA) and the middle cingulate cortex, extending to both the medial and the superior frontal gyri (Table 1). Importantly, the same pattern of results was also observed when *diff_bell* was considered, with however a higher degree of anatomo-functional specificity (Figure 3C and Table 1).”

- New Figure 3:

Figure 3: SVR-LSM results for (A) *line bisection* estimates, (B) *total_bell* and (C) *diff_bell*. The raw β -map is placed at the top, whereas the FDR-corrected (at $q = 0.05$) $(1-p)$ -map is underneath. The used hyper-parameters are displayed at the top. The prediction accuracy and reproducibility of the SVR-LSM models are indicated at the bottom. ad, antero-dorsal; GSM, supramarginal gyrus; pre-SMA, pre-supplementary motor area; SFG, superior frontal gyrus; SPL, superior parietal lobule. Note that $1-p$ values are used to facilitate visualization. See Table 1 for a detailed report of significant areas.

• New Figure 2:

Figure 2: Combined violin and notch boxplots of behavioral measurements. All individual data are shown. (A) The three assessments for the line bisection task. (B) Delta measures for the line bisection task. (C) The three assessments for *total_bell*. (D) Delta measures for *total_bell*. (E) The three assessments for *diff_bell*. (F) Delta measures for *diff_bell*. The center of boxplots represents the median value; the black lines, extreme values. * $p < 0.05$; ** $p < 0.01$; *** $p < 0.0001$ (see the main text for exact p -values). ns, non-significant. R software (<https://www.R-project.org/>; packages = *ggplot2* & *ggpubr*) was used to create this figure.

- New Table 1:

Table 1: SVR-LSM results for $\Delta 1$

AAL parcels	Significant voxels (n) in the parcel	Parcel percentage with significant voxels	Average p -values for significant voxels
Line bisection estimates (rightward deviations): $P_{FDR(q=0.05)} < 0.0028$			
Parietal_Sup	7979	45.5%	0.00051
Precuneus	3532	13.5%	0.00088
Angular	2933	20.9%	0.00103
Parietal_Inf	2325	21.6%	0.00124
SupraMarginal	3523	22.3%	0.00149
total_bell (total number of omitted bells): $P_{FDR(q=0.05)} < 0.004$			
Cingulum_Mid	5844	33.5%	0.00043
Supp_Motor_Area	7709	40.8%	0.00096
Frontal_Sup	7227	22.5%	0.00103
Frontal_Sup_Medial	2124	12.5%	0.00162
Parietal_Inf	1731	16.1%	0.00217
Parietal_Sup	2611	14.9%	0.00227
Angular	2417	17.3%	0.00310
diff_bell (left minus right bells): $P_{FDR(q=0.05)} < 0.003$			
Parietal_Sup	4999	28.5%	0.00038
Parietal_Inf	874	8.1%	0.00061
Cingulum_Mid	2980	17.1%	0.00063
Supp_Motor_Area	2881	15.3%	0.00105
Frontal_Sup_Medial	1054	6.20%	0.00121
Frontal_Sup	1707	5.3%	0.00127

The permutation-derived p -maps were thresholded with a FDR procedure (see Materials and Methods). Only areas harboring suprathresholded voxels in a proportion of at least 5% are detailed.

- New Figure 4:

Note that in the previous version of the figure, two errors occurred. First, the number of patients (n) was inversed between the parietal group *versus* the frontal group (not in the main text). Second, after careful checking during the revision, one patient with a SMA resection was omitted to be included in the MPC group. We apologize for this error. The related statistical analyses were thus performed again (only very minor changes), and those related to *diff_bell* added:

“Group analyses. To better highlight the dissociation described above, we directly contrasted the behavioral performances of all patients with a resection located in the parietal cortex ($n = 18$) *versus* located in the medial frontal lobe ($n = 21$) (Figure 4A). A two-way mixed ANOVA performed on the line bisection performance showed, as expected, a principal effect of *group* ($F_{(1, 37)} = 5.92$, $p = 0.02$, $\eta^2_p = 0.14$) and *assessment time* ($F_{(2, 74)} = 24.57$, $p = 6.53 \times 10^{-9}$, $\eta^2_p = 0.40$), and most importantly a **significant** interaction effect between both factors ($F_{(2, 74)} = 12.86$, $p = 0.000016$, $\eta^2_p = 0.26$). Post-hoc analyses revealed that performances differed between both groups, but only at A2 ($p = 0.00061$; $p > 0.10$ for comparisons at A1 and A3) (Figure 4B). With respect to *diff_bell*, a principal effect was found for *assessment time* ($F_{(2, 74)} = 29.29$, $p = 4.27 \times 10^{-10}$, $\eta^2_p = 0.44$) but not for *group* ($F_{(1, 37)} = 0.54$, $p = 0.47$, η^2_p

= 0.014). Both factors did not interact significantly ($F_{(2, 74)} = 0.69, p = 0.50, \eta^2_p = 0.018$) (Figure 4C). The same pattern of results was found for *total_bell* (see supplementary Figure 5).”

Figure 4: Results from group analyses. (A) Overlap maps for each group or patients. Bars indicate lesion density at each voxel. (B) Violin plot for line bisection performance. (C) Violin plot for *diff_bell*. ***, $p < 0.001$. See the main text for details about the statistical analyses.

Major comment #3: The image normalization procedure appears odd. You only report the use of “default normalization parameters” in SPM. However, as with all brain pathology visible in the scan, the pathology will interfere with the normalization process and potentially induce systematic errors (see e.g. PMC2938189; PMC2658465). Could you specify their procedure in detail? Note that cost function masking or enantiomorphic normalization will work not only on stroke lesions, but on tumors as well. If you did not use such strategy, I’d like you to provide some additional results to demonstrate good normalization. This could be, for example, a set of the first 15 patient images pre- and post-normalization.

The authors: We initially did not use cost function masking or enantiomorphic normalization because SPM12 performs very well without masking lesions during the registration process since its last major update (2017). Having said that, we acknowledge that this approach may be criticized given the previous literature which was however based on older versions of SPM. We thus decided to register again all native MRIs to MNI space using enantiomorphic

normalization (clinical toolbox). All tumour/resection cavities were consequently drawn again, and all analyses were performed again. The results remain the same. For information, we found that SPM12 normalization (without enantiomorphic step) underestimates lesion size in the case of large lesions. See the following changes:

- In the section ‘neuroanatomical data and lesion drawing’:

“ To minimize the potential bias caused by abnormal lesion-related radiological signals, MRI datasets were registered to the MNI space using enantiomorphic normalization⁶. This procedure was performed with the SPM12 (<https://www.fil.ion.ucl.ac.uk/spm/software/spm12/>) clinical toolbox (<https://github.com/neurolabusc/Clinical>)⁷. The output resolution was 1*1*4 mm for FLAIR images and 1-mm isometric for 3DT1 images. As a first step, the tumors/resection cavities were semi-automatically drawn using MRICron package (<https://github.com/neurolabusc/MRICron>) and further inflated by means of a three-dimensional smoothing procedure (3-mm full-width-at half-maximum [FWHM] Gaussian kernel with a threshold of 0.3). The obtained masks were then binarized and inserted during the registration process (during enantiomorphic normalization, the area covered by a particular mask is replaced by the undamaged homologous area within the contralesional hemisphere). Before proceeding further, all normalized MRIs were systematically and carefully checked to identify and potentially exclude inaccurate registrations. All were satisfactory at this stage. **Next, tumors and resections cavities were drawn again on the normalized MRIs**, yielding two three-dimensional volumes of interest (VOI) by patients. These VOIs were spatially smoothed with a 2-mm FWHM Gaussian kernel (threshold of 0.4). The whole procedure was performed by the same experimenter who shows highly-skilled expertise in neuro-anatomy (the first author).”

- Added references:

Nachev, P., Coulthard, E., Jäger, H. R., Kennard, C. & Husain, M. Enantiomorphic normalization of focally lesioned brains. *Neuroimage* **39**, 1215–1226 (2008).

Rorden, C., Bonilha, L., Fridriksson, J., Bender, B. & Karnath, H.-O. Age-specific CT and MRI templates for spatial normalization. *Neuroimage* **61**, 957–965 (2012).

Major comment #4: In the introduction, you claim that the multivariate “SVR-LSM [has] a greater biological value since the method takes into consideration the inter-dependent nature of the voxels forming a given lesion map and is in principle better suited to control for the non-stochastically spatial distribution of brain damage” This claim has been refuted by several previous studies (e.g. <https://doi.org/10.1016/j.neuropsychologia.2010.04.018>; <https://pubmed.ncbi.nlm.nih.gov/30549154/>, <https://onlinelibrary.wiley.com/doi/full/10.1002/hbm.25278>)

The authors: we modified the sentence as follows:

“**SVR-LSM represents an interesting option as the method takes into consideration the inter-dependent nature of the voxels forming a given lesion map.**”

Major comment #5: You used default hyperparameters for the SVR-LSM. This is per se a suboptimal procedure – we never know how well hyperparameters transfer from the sample in Zhang et al (who also used different parameters across analyses!) to your sample and deficit. This is even a more severe issue with the current sample, as hyperparameters were originally optimized on stroke data by Zhang et al.. However, given the tricky situation with model fit and reproducibility as opposing criteria in

SVR-LSM (Zhang et al., 2014), I could accept the use of the default parameters. In this case, I would like you to shortly mention this limitation and to report both values – out-of-sample model fit and reproducibility – for the chosen parameters. This is quite important to evaluate the quality of the SVR. Only if the SVR explains at least some behavioral variance, the mapping results can be considered reliable at all.

The authors: We thank the reviewer for this valuable comment. We agree on that. In the revised version of the manuscript, hyper-parameters of SVR-LSM have been optimized *via* grid search. As the reviewer will see, satisfactory parameters have been found for $\Delta 1$. Prediction accuracy and reproducibility of SVR-models are now reported. Please see the following changes:

- In the section ‘Materials and methods’:

“In this study, we used the Matlab script originally coded by Zhang et al.⁸ to perform all SVR-LSM analyses (<https://github.com/yongsheng-zhang/SVR-LSM>). *Epsilon*-SVR models with a radial basis kernel function (RBF) were used to estimate hyperplane. **Hyper-parameters of SVR-LSM models were optimized using a grid searching procedure, in particular the cost (C) which corresponds to the penalty/regularization parameter and gamma (γ) which represents the kernel coefficient. This optimization was performed by means of a 5-fold cross-validation procedure allowing to determine the combination of parameters that maximized prediction accuracy while maintaining a high-level of reproducibility (see Zhang et al. for a complete description of how model fit and reproducibility are estimated). In total, 66 couples of parameters were assessed for each behavioral measure of interest and for each time point i.e. A1, $\Delta 1$ and $\Delta 2$ (9 optimization procedures, sum-total), with $C = [1\ 10\ 20\ 30\ 45\ 50]$ and $\gamma = [0.1\ 1\ 2\ 3\ 4\ 5\ 6\ 7\ 8\ 9\ 10]$. Parameter assessment was done with a publicly available Matlab script (‘svr_lsm_BasicScript_opt.m’, <https://data.mendeley.com/datasets/2hyhk44zrj/2>)⁹ after it was checked for quality and modified for local use. Datasets associated with a poor prediction accuracy ($r_{max} < 0.20$) and/or an insufficient index of reproducibility ($r_{max} < 0.90$) were not eligible to subsequent SVR-LSM analyses.”**

- In the section ‘SVR-LSM’ results:

“**SVR-LSM results.** A grid searching approach was used to determine the optimal hyper-parameters (*i.e.* γ and C) of SVR-LSM models (see Materials and Methods). With respect to A1, we failed to identify a combination of hyper-parameters associated with a good prediction accuracy and a high level of reproducibility for the three measures of interest (See Supplementary Figure 3A for grid search). This was related to the fact that there was only a limited amount of pathological variance to be modeled given the little or the absence of behavioral differences between the control and the patient group at preoperative baseline (see Supplementary Figure 1 and 2) – in agreement with the established brain’s efficient abilities to reorganize in response to lower grade glioma progression³. By contrast, satisfactory hyper-parameters were identified for $\Delta 1$ for all measurements of visuo-spatial attention, including *line bisection* estimates ($\gamma = 4$, $C = 30$) *total_bell* ($\gamma = 5$, $C = 30$) and *diff_bell* ($\gamma = 5$, $C = 30$) (See Supplementary Figure 4 for grid search). The goodness-of-fit and reproducibility of selected models are displayed on Figure 3. The generated SVR-LSM maps [...]”

• Supplementary Figure 4:

Supplementary Figure 5: Grid search for SVR-LSM models of ΔI . The retained hyper-parameters are indicated on the figure. Note that the directionality of the x - and y -axis is reversed for prediction accuracy. The plots were created with Matlab's surface plot function.

• **Supplementary Figure 3:**

Supplementary Figure 4: Grid search for SVR-LSM models of $A1$ and $\Delta 2$. As shown in this figure, prediction accuracy and/or reproducibility were not enough to generate reliable lesion-symptom maps.

Note that the directionality of the x - and y -axis is reversed for prediction accuracy. The plots were created with Matlab's surface plot function.

Major comment #6: A main conclusion is that the common concept of a parietal correlate of line bisection and a frontal correlate of cancellation tasks are only part of the story, and the current study adds to it. However, I believe this summary of the current state to fall far too short. There are many studies that assessed neglect by cancellation and found parietal, temporal, and subcortical correlates. Of note, even Verdon et al. did not find exclusively frontal/exclusively parietal correlates, but always both to different degrees (see Figure 5). Given the issues with classic univariate VLSM that you mentioned, this might well be a methodological artifact. Please provide a more nuanced discussion of previous neglect studies (e.g. [10.1016/j.cortex.2011.11.008](https://doi.org/10.1016/j.cortex.2011.11.008); [10.1093/cercor/bhh076](https://doi.org/10.1093/cercor/bhh076); <https://doi.org/10.1016/j.neuropsychologia.2008.08.020>; [10.1093/brain/awg200](https://doi.org/10.1093/brain/awg200) and many more).

The authors: We thank the reviewer for this valuable comment. We initially did not extend the discussion on the vast literature about the neural correlates of spatial neglect because the format of *Nature Communications* is very restricting in terms of word count and reference limit as well. This is the reason why we give priority in our discussion to the results of meta-analyses that do not pool tasks together. For example, in the quantitative ALE-based meta-analysis by Molenberghs et al. (2011), line bisection was found to be more likely affected by posterior/parietal lesions, and cancellation tasks by frontal and in a much lesser extent posterior/parietal lesions. We acknowledge however that some individual studies have offered complementary results that go beyond this frontal/parietal dissociation.

In the discussion section, we made the following changes and added two references suggested by the reviewer:

"[...] that has been gaining attraction over the years, but not fully evidenced by experimental data, is that perceptual tasks (such as the line bisection task) may be preferentially affected by lesions damaging the posterior parietal cortex, whereas visuo-motor exploratory tasks (such as the bell test) may rather be affected by lesions targeting the frontal lobe^{1,43,44,46} – **knowing that this general pattern may be in reality more complex as cancellation performances have been also found to be impaired following posterior or subcortical lesions (e.g. ^{56,57}).**

Added references

Karnath, H.-O., Fruhmann Berger, M., Küker, W. & Rorden, C. The anatomy of spatial neglect based on voxelwise statistical analysis: a study of 140 patients. *Cerebral Cortex* **14**, 1164–1172 (2004).

Mort, D. J. *et al.* The anatomy of visual neglect. *Brain* **126**, 1986–1997 (2003).

Major comment #7: I have some doubt about the single patient. If I understood correctly, you did the same disconnection analysis in the LQT Toolbox for this single patient. If I remember correctly, the toolbox uses probabilistic maps of WM tracts. This might not be problematic with large samples, but a strong conclusion from a single case is on thin ice. You should carefully discuss what kind of data the analysis is based on (i.e. is it really probabilistic?) and discuss this as a limitation given considerable variance in the anatomy of fiber tracts. Regarding this variance, see e.g. the atlas by Zhang et al ([10.1016/j.neuroimage.2010.05.049](https://doi.org/10.1016/j.neuroimage.2010.05.049)), where the frontoparietal part of the SLF is in no voxel more likely to be present than ~80%, hence we can hardly ever guarantee affection of the SLF-FP in any single patient.

The authors: We confirm that we performed the same disconnection analysis in the single patient than that made at the group level. This atlas is not “probabilistic” strictly speaking in

that the tracts are generated from the averaged diffusion data of the 1065 HCP subjects. The clear advantage is to deal with tracts that represent the most commonly shared features of white matter architecture at the population level and that are constructed on the basis of an impressive set of individual data. The disadvantage is to lose information related to the inter-individual variability (this is in some extent counterbalanced by the number of subjects used to reconstruct the fiber tracts that represent the most commonly shared spatial arrangements). The output results give a percentage of damaged fibers which is in our opinion a much better estimate of tract disconnection than those commonly used in studies such as tract load or tract damage probabilities. Note that more recent atlas found commonly shared voxels across all subjects for SLF_I (Rojkova et al., 2016, is just an example among others).

Having said that, we acknowledge that the statement related to SLF_I sounds a little bit strong given that tract inter-individual variability is not directly taken into consideration in the LQT analyses. So, we agree to tone down it.

Note however that our interpretation remains valid because if SLF_I disconnection was actually associated with line bisection deficits, then patients of the MFC would have shown deficits - the cortical projections of SLF_I mainly targeting SMA, pre-SMA, anterior/middle cingulate and medial SFG (widely damaged in the MFC group).

- In the section 'discussion':

"This conclusion is bolstered by the observation that **the likely interruption** of the fibers forming the SLF_I and the cingulum did not result in rightward deviations in the patient with a sequential surgery (**see Supplementary Note 1**).

- In the supplementary information:

Supplementary Note 1

"In the discussion section of the article, we mentioned that the fibers of SLF_1 were likely to be damaged in the single patient after the first surgery, while the disconnection analysis indicated that the fibers were almost completely interrupted. We interpreted this result with caution in view of the method used to estimate disconnection severity. The measures of disconnection computed by the LQT toolbox is based on fiber tracts generated from the averaged diffusion data of the 1065 HCP participants. As comprehensively discussed by Griffis et al.¹, the clear advantage is to deal with tracts that represent the most commonly shared features of white matter architecture at the population level and that are constructed on the basis of an unprecedented sample of individual data. The counterpart is that the interindividual variability in the spatial arrangement of tracts are not directly taken into consideration in the disconnection analyses. As a consequence, if the SLF_I of the single patient has an "outlier" spatial distribution, we cannot formally exclude the possibility that the tract is less affected in reality. While the disconnection results of the single patient does not allow to provide strong conclusion about the role of this tract in line bisection deficits, this does not alter the suitability of the interpretation we proposed. If damage to SLF_I is a central mechanism underlying the occurrence of line bisection deficits, then such deficits are expected to vastly occur in the MPF group because the cortical projections of SLF_I mainly target SMA, pre-SMA, anterior/middle cingulate and medial SFG²⁻⁴. Our results did not reveal such a behavioral pattern".

1. Griffis, J. C., Metcalf, N. V., Corbetta, M. & Shulman, G. L. Lesion Quantification Toolkit: A MATLAB software tool for estimating grey matter damage and white matter disconnections in patients with focal brain lesions. *BioRxiv* (2020).
2. Makris, N. *et al.* Segmentation of subcomponents within the superior longitudinal fascicle in humans: a quantitative, in vivo, DT-MRI study. *Cerebral Cortex* **15**, 854–869 (2005).

3. Rojkova, K. *et al.* Atlasing the frontal lobe connections and their variability due to age and education: a spherical deconvolution tractography study. *Brain Structure and Function* **221**, 1751–1766 (2016).
4. Komaitis, S. *et al.* Dorsal component of the superior longitudinal fasciculus revisited: novel insights from a focused fiber dissection study. *Journal of Neurosurgery* **132**, 1265–1278 (2019).

Minor comment #1: I don't really understand the purpose of Background analyses, the first section of the results. You state that you wanted "to gauge the weight of [demographic] variables on the behavioral measurements of visuo-spatial attention." What is the 'weight' of a demographic variable? And why do you include tumor size/resection volume in the analysis? In the SVR-LSM, you perform a regression of brain damage location (not size!) on symptoms. Hence, it doesn't look to me like you mean it serious with the factor tumor size/resection volume. Maybe age just explains variance in your analysis because resection volume isn't such a good predictor, and voxel-wise (/tract-wise) anatomical data would explain the data, leaving nothing to age. I would prefer a simple descriptive analysis.

The authors: As we did not use the clinical and demographic variable as covariates of non-interest in the lesion-symptom analyses, we had previously thought that it was important to show that these variables did not (or poorly) correlated with the behavioral measurements. We have reworded the section in question in a more descriptive way, as suggested. For the sake of completeness, we provide a correlation table in the supplementary data.

- In the main text:

“Patient sample and background analyses. Background demographic and clinical variables are fully described in Supplementary Table 1. In brief, the patient sample consisted of 128 patients (mean age: 39.7 ± 12.3 , 54 females; 121 right-handed) consecutively operated on for a lower grade glioma (see Materials and Methods for details about inclusion and exclusion criteria). They were behaviorally assessed at three time points: the day before surgery (hereafter, A1), four day after surgery (hereafter, A2) and three months after surgery (hereafter, A3). The average preoperative volume of tumors was $57.5 \text{ cm}^3 \pm 49.0$, whereas the average volume of postoperative resection cavities was $47.2 \text{ cm}^3 \pm 39.7$. Simple correlation analyses indicated that the behavioral measurements of visuo-spatial attention, including *line bisection* estimates, the total number of omitted bells (Hereafter, *total_bell*) and the asymmetry score left minus right bells (Hereafter, *diff_bell*), were poorly associated with the demographic and clinical variables (see Supplementary Table 2). This was true for the preoperative level of performance (A1) but also for $\Delta 1$ (*i.e.* the behavioral difference between A1 and A2) and $\Delta 2$ (*i.e.* the behavioral difference between A1 and A3). As a consequence, the variance associated with these variables were not regressed out from the behavioral measures of interest in the subsequent lesion-symptom analyses.”

- In Supplementary information:

Supplementary Table 2: Correlation analyses between the behavioral measurements of visuo-spatial attention and the demographic/clinical data

		age	education	lesion volume*
line bisection	A1	-0.02	-0.12	0.01
	$\Delta 1$	-0.08	0.00	-0.11
	$\Delta 2$	-0.09	0.00	0.13
total_bell	A1	-0.06	-0.06	0.14
	$\Delta 1$	0.13	0.07	-0.02
	$\Delta 2$	0.25	0.002	0.16
diff_bell	A1	0.00	0.05	-0.10
	$\Delta 1$	-0.05	0.05	-0.13
	$\Delta 2$	-0.16	0.16	0.05

Only one slight but significant correlation (in bold) was found between *age* and *total_bell* ($\Delta 2$).

* Lesion volume corresponds to preoperative tumor volumes for A1 and to postoperative resection cavity volumes for A2, A3, delta1 and delta2.

Minor comment #2: In the section “patient sample”, a small side remark states that no patients suffered from visual extinction. This is quite interesting, given that some resections included parietal areas. How and when was extinction tested? It would be great if you could provide some more information on this point.

The authors: Visual extinction was assessed at each time point (i.e. before surgery, four days after and three months after surgery) with a double simultaneous stimulation test (performed with fingers). We have made this clarification in the current version of the manuscript.

- Section ‘patient sample’:

“The patients’ sociodemographic and clinical characteristics are given in Supplementary Table 1. **Note that none of the patients suffered from visual extinction before, a few days after and three months after surgery. It was assessed using a double simultaneous stimulation test (i.e. visual stimulations were performed with a finger either unilaterally, left or right, or bilaterally).**”

Minor comment #3: Regarding correction for multiple comparisons, you state that “A FDR correction ($q = 0.05$) estimated by a bootstrap permutation procedure (5000 permutations) was used to threshold the resulting SVR-LSM β -maps”. This is quite imprecise. There is first a permutation to estimate the significance of the voxel’s feature weight, and then FDR is applied. Anyway: Were all

methods applied inside the SVRLSM scripts from Zhang et al? Please state. Could you cite the applied FDR method?

The authors: Thank you for your careful reading; this sentence is indeed inaccurate. We furthermore confirm that the used FDR procedure is that implemented in the Zhang's original script and described in the Nichols's paper (Genovese, Lazard and Nichols; *Neuroimage* 2002).

[...] for lesion volume directly to the lesion data (and not to the behavior performances). A bootstrap permutation procedure (5000 permutations) was first used to estimate the significance of the voxels' feature weight. Then, a FDR correction¹⁶ ($q = 0.05$) (directly implemented in the script by Zhang et al.⁸) was applied to threshold the resulting SVR-LSM p -maps.

Added reference

Genovese, C. R., Lazar, N. A. & Nichols, T. Thresholding of statistical maps in functional neuroimaging using the false discovery rate. *Neuroimage* **15**, 870–878 (2002).

Minor comment #4: Group analyses: the ANOVA showed “most importantly a powerful interaction“. The term ‘powerful’ is imprecise here. It’s highly significant, that’s all. As the term ‘statistical power’ exists, this might be confusing.

The authors: We agree. “Powerful” has been replaced by “significant.”

Minor comment #5: In the discussion, you interpret p -values from correlation analysis by their magnitude: some “were clearly more significant”. Aside from the problems with a comparison of significance levels in inferential statistics, why don’t you just look at the correlation values, which tell you something about the actual strength of a relation between two variables?

The authors: We agree. We modified the sentence as follows:

“[...] the correlations observed for the two latter tracts were greater for total_bell/diff_bell vs. line bisection - in line with our prior hypotheses.”

Reviewer #3

This is a potentially interesting paper: nonetheless some clarifications and rewriting seems mandatory at the moment. In the following I have detailed a number of points that the authors might wish addressing in revision.

The authors: We thank the reviewer for his/her critical review of the work. We have followed all reviewer’s suggestions.

Comment #1: Intro page 3: please note that there is growing evidence that a strict dorsa-ventral distinction between endogenous/orienting and exogenous/re-orienting control, respectively, might be

incorrect because in response to invalid targets: 1) the ventral TPJ area is in charge of the late processing of the match/mismatch between the cued and actual target location for updating the contextual probabilistic association between cue direction and target location (see Doricchi et al., 2010; Geng and Vossel, 2013; Macaluso and Doricchi, 2013); b) the early operation of re-computing the direction of attention to the actual target location, i.e. spatial reorienting, is played by the superior rather than inferior parietal lobe (Vandenberghe et al., 2012; Ptak and Schneider, 2011): note that the findings reported by the authors support this view. The authors might wish to smooth down the old-fashioned dorsal/ventral dichotomy.

The authors: We thank the reviewer for this suggestion. In the introduction section, we now toned down the strict ventral/dorsal dichotomy. Note that we are very limited in the permitted number of references. So, we only cite two of the suggested references. Please see the change:

“In particular, the dorsal attention network (DAN), mainly composed of the frontal eye field (FEF) and the intraparietal sulcus, may subserves the ability to purposely allocate attention to meaningful elements of the visual scene (top-down, goal-directed orientation), whereas the ventral attention network (VAN), composed of the ventro-lateral prefrontal cortex and the temporo-parietal junction, may be engaged when an unexpected but behaviorally relevant event occurs and attention must be reoriented towards the new visual target (bottom-up, stimuli-driven orientation) – **though this dual-pathway anatomofunctional organization would be less marked than previously thought**^{17,18}.

● Added references:

Macaluso, E. & Doricchi, F. Attention and predictions: control of spatial attention beyond the endogenous-exogenous dichotomy. *Frontiers in Human Neuroscience* **7**, 685 (2013).

Geng, J. J. & Vossel, S. Re-evaluating the role of TPJ in attentional control: contextual updating? *Neuroscience & Biobehavioral Reviews* **37**, 2608–2620 (2013).

Comment #2: Page 2 Intro: “However, almost nothing is known about the contribution of these medial areas ...” may be it would be better that “ little is known “ (see for example ref. 56, 57). Same when later on the authors state “...neuropsychological studies have consequently failed in delineating the exact role of the dorsomedial structures “

The authors: We have toned down the sentences pointed by the reviewer, as follows:

- “However, **little is** known about the contribution of these medial areas to the voluntary control of visual attention, especially in humans.”

- “As the dominating lesion model has been invariably stroke injury, neuropsychological studies **have faced difficulty in delineating the** exact role of the dorsomedial structures [...]

Comment #3: Page 5 Intro: “While the association between SLF II and III and spatial neglect has been experimentally evidenced^{25,26,49}, the causal role of SLF I remains elusive.... “, at least partially elusive: ref 56 reported the effect of SLF I disconnection.

The authors: We have modified the sentence in question, as follows:

“While the association between SLF II and III and spatial neglect has been experimentally evidenced^{4,19,20}, the exact contribution of SLF_I remains to be clearly elucidated.”

Comment #4: Page 11: “In summary, the above analyses confirmed that the bell test was affected to the same extent by resections targeting either the parietal or the fronto-medial areas. By contrast, line bisection performances were uniquely impaired following parietal resections. This means that resections of the medial frontal lobe (especially SMA and the adjacent middle/anterior cingulate cortex) specifically impacted left_bell.” I’m not fully convinced by the logic of this sentence: these results suggest that parietal resections specifically impacted line bisection whereas both parietal and medial frontal lobe resections impacted left bell. The authors might wish to rewrite or drop this paragraph.

The authors: In the revised version of the manuscript, we have suppressed this sentence.

Comment #4: Page 17: “Our findings clearly demonstrate that damage to these two eye-related areas is able to produce visuo-motor exploratory neglect, though the underlying mechanisms cannot be fully elucidated here; it might be either the consequence of a deficit in initiating saccades towards the contralesional space, in line with the established behavioral affiliations of the medial frontal structures⁵⁷, or the result of specific difficulties in deliberately orienting attention during visual search (i.e. the attentional function attributed to the FEF). The latter interpretation might be more likely, as ablation or inactivation studies in monkeys have generally shown not or only little impact of SEF/CEF damage on saccade initiation^{54,58}. Another possibility is that visuo-motor exploratory neglect arises from a functional diaschisis effect⁵⁹ transiently impairing the FEF by deprivation of functional inputs – the FEF, SEF and CEF being densely interconnected^{19,20}. “ On the one hand I’m very positively impressed by the scholarly compilation of such an exhaustive set of hypotheses though, on the other hand, some of these hypotheses weaken significantly the strength of the main conclusion of this paper, i.e. that it provides first evidence for a role of the frontal-medial network in the voluntary deployment of visuo- spatial attention. Put in other words, although the study is methodologically sound and was scholarly run on an extended samples of carefully clinically selected patients with low-grade brain gliomas, the same study looks like providing new interesting insights in the neglect syndrome though not final evidence on the role played by the frontal-medial network. I think the authors might wish to “positively” smooth the tone of their conclusions and treat the empirical evidence they have gathered as importantly pointing at the possible role of medial frontal-parietal structures in the guide of attention.

The authors: We thank the reviewer for his/her comment. We realize that the hypotheses developed in this paragraph were not enough “hierarchically” organized, probably to avoid any overstatement. Of course, the most likely hypothesis is that the medial eye field network is important for the deployment of visuospatial attention, given the neuropsychological data we provided. Please see the following changes:

- In the paragraph in question:

“[...] Our findings clearly demonstrate that damage to these two eye-related areas is able to produce visuo-motor exploratory neglect, though the underlying mechanisms cannot be fully elucidated here; it might be either the consequence of a deficit in initiating saccades towards the contralesional space, in line with the established behavioral affiliations of the medial frontal structures²¹, or the result of specific difficulties in deliberately orienting attention during visual search (i.e. the attentional function attributed to the FEF). The latter interpretation is more likely since ablation or inactivation studies in monkeys

have generally shown not or only little impact of SEF/CEF damage on saccade initiation^{22,23}. Finally, we cannot fully exclude the possibility that visuo-motor exploratory neglect partially arises from a functional diaschisis effect²⁴ transiently impairing the FEF by deprivation of functional inputs – the FEF, SEF and CEF being densely interconnected^{25,26}.

- In the conclusion:

“In conclusion, our findings provide new evidence for an important role of the medial eye field network in the voluntary deployment of attention, as its disruption causes unilateral visuo-motor exploratory neglect. These findings have important implications for current neurocognitive and neurocomputational models of visuo-spatial attention.

REVIEWER COMMENTS

Reviewer #1 (Remarks to the Author):

The authors have made extensive and considered revisions addressing all main comments and suggestions. The additional analyses clarify some of the interpretational ambiguities and the revised manuscript is transparent about some inevitable limitations. The authors have carefully considered all the suggestions and potential impact on their initial conclusions. There remains scope for some academic difference in interpretation around how much can be concluded in relation to some of the hypotheses (e.g. around local disconnection vs network effects) which are thought-provoking and should stimulate further research. I have no further questions and have no further reservations in relation to publication of the manuscript.

Reviewer #2 (Remarks to the Author):

Dear authors,
this was an outstanding revision! You even did the hyperparameter optimisation although I haven't asked for it explicitly.

All my comments were satisfyingly addressed, the studies limitations are now clearly stated, and additional results & methods now provide more exact insights into the study's methods.

Minor

- typo in results "four day after surgery"

Reviewer #3 (Remarks to the Author):

In their review the authors have well addressed the points I have made in my review of this interesting study.

Response to Reviewers

Reviewer #1 (Remarks to the Author):

The authors have made extensive and considered revisions addressing all main comments and suggestions. The additional analyses clarify some of the interpretational ambiguities and the revised manuscript is transparent about some inevitable limitations. The authors have carefully considered all the suggestions and potential impact on their initial conclusions. There remains scope for some academic difference in interpretation around how much can be concluded in relation to some of the hypotheses (e.g. around local disconnection vs network effects) which are thought-provoking and should stimulate further research. I have no further questions and have no further reservations in relation to publication of the manuscript.

The authors: Thank you very much!

Reviewer #2 (Remarks to the Author):

Dear authors,
this was an outstanding revision! You even did the hyperparameter optimisation although I haven't asked for it explicitly.

All my comments were satisfyingly addressed, the studies limitations are now clearly stated, and additional results & methods now provide more exact insights into the study's methods.

Minor

- typo in results "four day after surgery"

The authors: Thank you again for your methodological advises. The typo is corrected.

Reviewer #3 (Remarks to the Author):

In their review the authors have well addressed the points I have made in my review of this interesting study.

The authors: Thank you again for your time in reviewing the manuscript.